# RETHINKING THE VALUE OF MULTI-AGENT WORK-FLOW: A STRONG SINGLE AGENT BASELINE

## ABSTRACT

Recent advances in LLM-based multi-agent systems (MAS) show that workflows composed of multiple LLM agents with distinct roles, tools, and communication patterns can outperform single-LLM baselines on complex tasks. However, most frameworks are homogeneous, where all agents share the same base LLM and differ only in prompts, tools, and positions in the workflow. This raises the question of whether such workflows can be simulated by a single agent through multi-turn conversations. We investigate this across seven benchmarks spanning coding, mathematics, general question answering, domain-specific reasoning, and real-world planning and tool use. Our results show that a single agent can reach the performance of homogeneous workflows with an efficiency advantage from KV cache reuse, and can even match the performance of an automatically optimized heterogeneous workflow. Building on this finding, we propose **OneFlow**, an algorithm that automatically tailors workflows for single-agent execution, reducing inference costs compared to existing automatic multi-agent design frameworks without trading off accuracy. These results position the single-LLM implementation of multi-agent workflows as a strong baseline for MAS research. We also note that single-LLM methods cannot capture heterogeneous workflows due to the lack of KV cache sharing across different LLMs, highlighting future opportunities in developing *truly* heterogeneous multi-agent systems.

## 1 INTRODUCTION

Recent advances in large language models (LLMs) have sparked significant interest in multi-agent systems (MAS), where multiple LLM agents collaborate through predefined workflows to tackle complex tasks. These systems typically consist of specialized LLM agents, each defined by distinct system prompts and tools, that communicate according to specific patterns to achieve superior performance compared to single-LLM approaches (Zhuge et al., 2024; Liu et al., 2024; Zhang et al., 2025e). Current research has demonstrated the effectiveness of such multi-agent workflows across diverse domains, from mathematical reasoning to code generation and tool usage (Hu et al., 2025; Zhang et al., 2025b; Wang et al., 2025b).

However, a critical observation about existing MAS reveals a fundamental characteristic that has been largely overlooked: **most current multi-agent systems are homogeneous** (Ye et al., 2025a; Zhang et al., 2025a). Within a given MAS, all agents rely on the same base LLM, differentiated only by their system prompts, tools, and positions in the workflow. This homogeneity raises a compelling question: if all agents share the same underlying model and work **collaboratively** to solve a task, can a single agent simulate the multi-agent workflow effectively through multi-turn conversations?

It is well recognized that task decomposition is critical for solving complex problems, which directly motivates the design of multi-agent systems. Given a multi-agent workflow that already decomposes a task, what happens if a single agent executes the workflow end-to-end? Specifically, because homogeneous agents possess identical reasoning capabilities and differ only in their specialized instructions, a single agent should be capable of role-playing these agents sequentially, thereby exploiting the workflow's task decomposition. Moreover, a single agent can reuse a shared KV cache across agent interactions, retaining context without additional prefill cost and potentially offering efficiency gains and greater consistency than maintaining separate model instances for each agent.

To investigate this hypothesis, we conduct comprehensive experiments across seven benchmarks spanning coding, mathematics, general question answering, domain-specific reasoning, and real-world planning and tool use. Our results reveal that *a single agent using multi-turn conversations with KV cache can indeed simulate tailored workflows with performance comparable to traditional homogeneous multi-agent setups, while reducing cost.* This finding challenges the conventional assumption that multiple separate agent instances are necessary for effective tailored reasoning.

Building on this insight, we introduce **OneFlow**, an algorithm for automatically designing tailored workflows that optimize both performance and computational efficiency for single-agent execution. Based on recent work on automatic workflow design (Zhang et al., 2025e;b; Nie et al., 2025; Hu et al., 2025), OneFlow employs a dual-LLM designer architecture to discover streamlined workflows with longer, more comprehensive system prompts for individual agents and fewer total agents in the system. This approach achieves similar performance to existing methods while significantly reducing inference cost.

We further extend our analysis to heterogeneous multi-agent workflows, where agents use different base LLMs (Ye et al., 2025a; Zhang et al., 2025a; Wang et al., 2024a). Exhaustively exploring combinations of base models can be very expensive (Ye et al., 2025a). In a pilot experiment using AFlow (Zhang et al., 2025e) to automatically design heterogeneous workflows via Monte Carlo Tree Search, we find that our single-LLM baseline can match the performance of one such automatically discovered heterogeneous alternative with less computational cost. Importantly, single-LLM approaches have inherent limitations: they cannot simulate *truly* heterogeneous workflows due to the inability to share KV caches across different models. This limitation suggests that developing effective heterogeneous agentic workflows, where the benefits of model diversity outweigh coordination costs, remains a promising and necessary direction for advancing LLM multi-agent system research. Our contributions can be summarized as follows:

- Empirically validate that single-agent execution can effectively simulate **homogeneous** multi-agent workflows with comparable performance on collaborative tasks.
- Propose **OneFlow**, an algorithm for automatic workflow design that generates streamlined multi-agent architectures with improved computational efficiency and suitability for single-agent execution.
- Show in a pilot study that a single-LLM implementation can match the performance of one automatically discovered heterogeneous workflow; more importantly, realizing the full benefits of heterogeneity remains an important open direction for agentic LLM systems.

## 2 PRELIMINARY

**LLM-based Multi-Agent Workflows.** Multi-agent workflows represent a paradigm where multiple LLM agents collaborate through structured communication patterns to solve complex tasks. To illustrate this concept, consider the session-based query recommendation example shown in Figure 1. Given a customer's shopping history and current query, the task requires understanding session context, analyzing product relationships, and generating relevant recommendations, a multi-faceted problem that benefits from specialized processing stages, as demonstrated in the middle panel of Figure 1. Importantly, these workflows are implemented as executable Python code (right panel of Figure 1) that specifies both the agents and their interaction logic. This code-based representation enables complex control flows including sequential execution, conditional branching, iterative refinement, and collaborative deliberation patterns.

**Formal Definition.** We now provide a formal characterization of LLM-based multi-agent workflows. Let $\mathcal{M}$ denote the set of available base LLMs. An LLM-based multi-agent workflow $W$ is defined as a directed graph $G = (N, E)$ where:

- $N = \{n_1, n_2, ..., n_{|N|}\}$ represents the set of LLM agents. Each agent $n_i$ is parameterized as $n_i = (b_i, p_i, \tau_i)$, where: $b_i \in \mathcal{M}$ is the base LLM (e.g., Claude 3.5 Haiku, Gemini 2.5 Flash), $p_i$ is the system prompt that defines the agent's role and capabilities, $\tau_i$ is the available tool set (e.g., sandboxed Python interpreter, web search). Agents are typically specialized for specific subtasks, with common roles including LLM reviewers, LLM ensemblers, and output formatters.

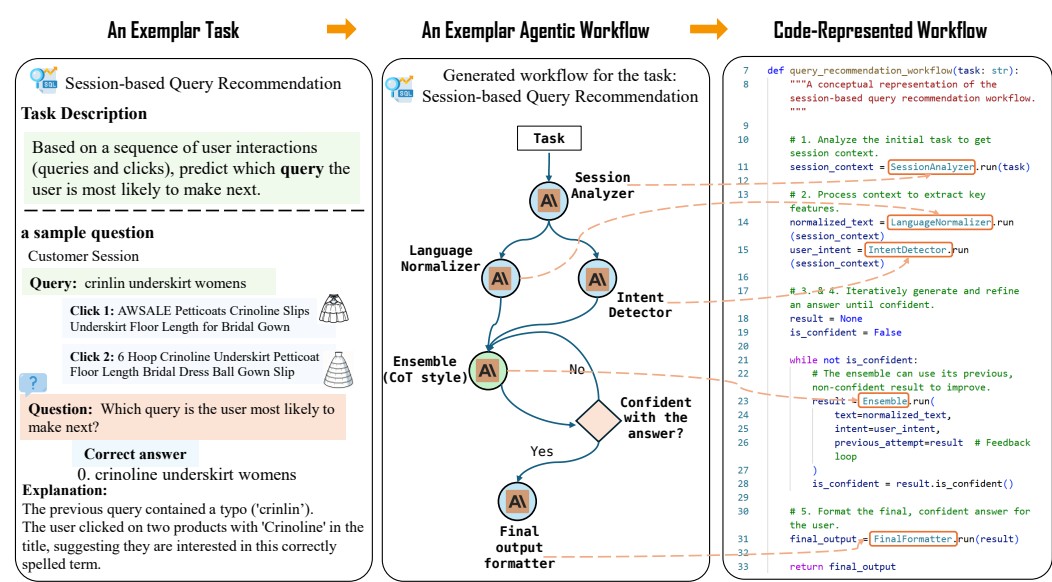

Figure 1: Sample question–answer pair from a session-based query recommendation task (left), an exemplar agentic workflow to solve it (middle), and its code representation (right). The workflow demonstrates how multiple LLM agents can collaborate to process complex shopping queries through sequential and conditional execution patterns.

- $E \subseteq N \times N$ encodes the inter-agent communication structure and control flow. Each edge may include routing conditions and message transformations implemented in Python, enabling sophisticated orchestration patterns such as sequential processing, conditional branching, and iterative loops.

**Homogeneous vs. Heterogeneous Workflows.** A critical distinction in multi-agent workflows concerns the diversity of underlying base models. We define $\mathcal{B}(W) = \{b_i \mid n_i \in N\}$ as the set of base LLMs used by workflow $W$. Based on this, workflows can be categorized as:

- **Homogeneous workflows**: $|\mathcal{B}(W)| = 1$, where all agents share the same base LLM and differ only in their system prompts, tools, and positions within the workflow structure.
- **Heterogeneous workflows**: $|\mathcal{B}(W)| > 1$, where agents utilize different base LLMs, potentially leveraging diverse model capabilities and specializations.

*Design Complexity of heterogeneous workflows.* The choice of which base model to assign to each agent (the mapping $i \mapsto b_i$) represents a non-trivial design decision that is often determined empirically. While heterogeneous workflows offer greater design flexibility by combining models with complementary strengths, they also significantly expand the design space to include model selection alongside prompt engineering, tool assignment, and routing logic (models × prompts × tools × routing). Consequently, many existing automatic workflow design systems default to homogeneous configurations for practical reasons.

**KV Cache.** The distinction between **heterogeneous** and **homogeneous** workflows is fundamental to our analysis, as it determines whether a workflow can be efficiently simulated by a single agent instance through multi-turn conversations with shared KV cache. In transformer-based LLMs, the key-value (KV) cache is a crucial optimization technique that stores the computed key and value matrices from attention layers for previously processed tokens. Without KV caching, the model would redundantly recompute these attention states for all previous tokens when generating each new token, leading to quadratic computational complexity. By caching these intermediate states, the model achieves significant speedup during autoregressive generation. In **homogeneous** multi-agent workflows, where all agents share the same base LLM, there exists substantial contextual overlap between agent interactions, such as shared task descriptions, intermediate reasoning steps,

and common knowledge bases. This overlap enables efficient KV cache sharing across different agent roles within a single LLM instance, potentially offering both computational efficiency gains and improved consistency compared to maintaining separate model instances for each agent.

# 3 METHODOLOGY

## 3.1 SINGLE AGENT CAN BE AS STRONG AS MULTI-AGENT FRAMEWORK.

We formalize when a single agent can implement a homogeneous multi-agent workflow without loss of expressivity. Recall the formalization in the Preliminary section: a workflow $W$ is a directed graph $G = (N, E)$ with agents $n_i = (b_i, p_i, \tau_i)$ and routing logic on edges. Let $W$ be homogeneous with $|\mathcal{B}(W)| = 1$ and base LLM $b$. Executing $W$ on input $x$ produces a transcript

$$H_T = (h_0, m_1, r_1, h_1, \ldots, m_T, r_T, h_T),$$

where at step $t$ the workflow selects agent index $i_t$ and tool action $a_t$ according to a policy $\pi_W(i_t, a_t \mid h_{t-1})$ induced by $E$, queries the same base model $b$ with system prompt $p_{i_t}$ and context $h_{t-1}$ to obtain model message $m_t$, optionally executes a tool $u_t \in \tau_{i_t}$ to obtain result $r_t$, and updates the history $h_t$.

Consider a single-LLM simulator that maintains one conversation state and, at each step $t$, sets the system message to $p_{i_t}$, appends the same visible context $h_{t-1}$ and tool outputs, and decodes from the same base model $b$ with identical decoding parameters.

**Proposition 1 (Simulation of homogeneous workflows).** Suppose (i) tool side-effects are deterministic given inputs, (ii) the routing policy $\pi_W$ depends only on the visible history $h_{t-1}$ and tool outputs, and (iii) decoding uses deterministic rules (e.g., greedy) or shared randomness. Then the single-LLM simulator induces the same distribution over transcripts as executing $W$ with separate agent instances:

$$H_T^{\text{single}} \stackrel{d}{=} H_T^{\text{multi}}.$$

*Proof sketch.* Both procedures query the same conditional distribution $b(\cdot \mid p_{i_t}, h_{t-1})$ at the same sequence of states; induction on $t$ yields equality in distribution.

**Cost with KV cache.** Let $L_t$ be the tokenized length of the prefix visible at step $t$ and $\Delta L_t$ the number of new tokens appended between steps $t-1$ and $t$. With separate agent instances (no cache sharing), overlapping prefixes are re-encoded, yielding cost

$$C_{\text{multi}} \propto \sum_{t=1}^{T} L_t^{(i_t)} + \text{gen}_t.$$

The single-LLM simulator reuses the same KV cache across $t$, so

$$C_{\text{single}} \propto \sum_{t=1}^{T} \Delta L_t + \text{gen}_t \ \leq \ C_{\text{multi}},$$

with equality only when agent contexts are disjoint. Thus, for homogeneous workflows with substantial contextual overlap, single-LLM execution is asymptotically no worse and often cheaper, while preserving behavior under the conditions above.

## 3.2 SINGLE AGENT IMPLEMENTATION OF MULTI-AGENT WORKFLOW.

We provide a concrete single-LLM simulator for any homogeneous workflow $W = (N, E)$ with base LLM $b$. Let $n_i = (b, p_i, \tau_i)$. The simulator maintains a single chat history $h_t$ and follows the routing policy encoded by $E$.

1. **Initialization.** Set $h_0 = \text{wrap}(x)$ with task $x$ and global instructions. Insert role delimiters to isolate subsequent agent turns.

2. **Agent step $t$.** Using the routing logic in $E$, select index $i_t$ and required tool action $a_t$ based on $h_{t-1}$. We append the system message $p_{i_t}$ to the end of the conversation history $h_{t-1}$ (effectively treating the 'system message' as a user message), keeping all previous context. Then, query $b$ to obtain $m_t \sim b(\cdot \mid p_{i_t}, h_{t-1})$ with fixed decoding parameters.

3. **Tool execution.** If $a_t$ invokes a tool $u \in \tau_{i_t}$, execute $u$ to get result $r_t$ and append it to the history.

4. **State update.** Set $h_t = \text{append}(h_{t-1}, (i_t, m_t, r_t))$ and advance according to edge conditions in $E$. Repeat until termination.

Because every step calls the same base model $b$, the simulator reuses the KV cache across $t$ (see Preliminary), so the prefill cost scales with incremental growth $\Delta L_t$ rather than the full prefix $L_t$. To control context growth and mitigate interference, one may insert compaction operators (e.g., deterministic summarization) that map a window of turns $(h_{t-k:t}) \mapsto s_t$ where $s_t$ is the summarized representation, while preserving routing decisions, which leaves Proposition 1 applicable.

## 3.3 Algorithm: OneFlow

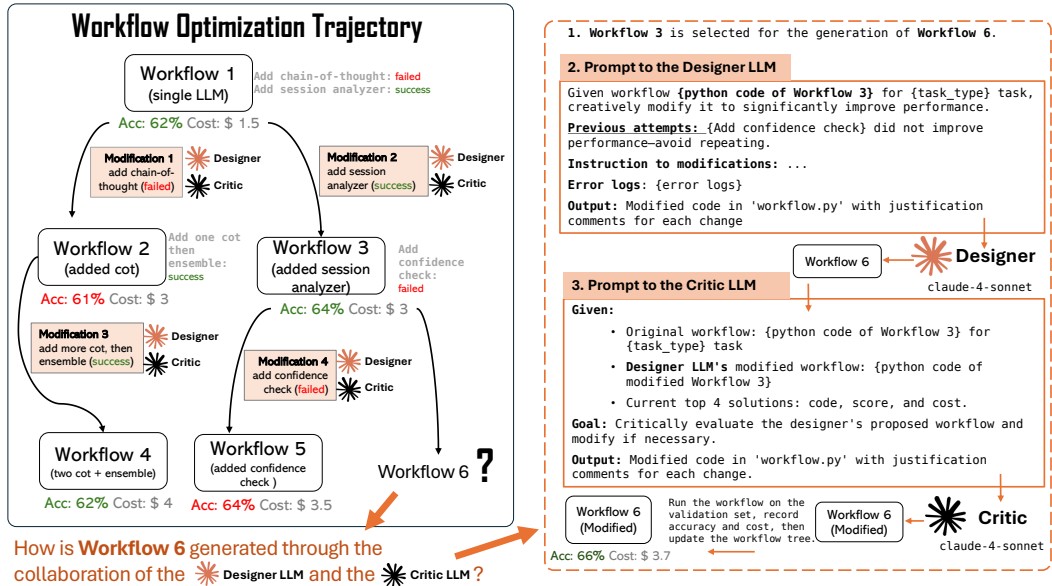

Figure 2: An example to show how OneFlow framework works. The framework employs dual meta-LLMs (One creative workflow designer and one workflow critic) with Monte Carlo Tree Search to automatically design multi-agent workflows that suitable for single-agent execution for complex tasks. The left panel shows how the first five rounds of workflow design and selection process work. The right panel shows how workflow 6 is generated.

For a given task $T$ (e.g., select a most relevant query to the customer from four candidate queries), we seek to return an optimal LLM agentic workflow $W^*$ that is suitable for this task $T$. Ideally, the optimal workflow $W^*$ should be able to solve the task $T$ with high performance and low cost. We formulate this as a multi-objective optimization problem:

$$W^* = \arg \max_{W \in \mathcal{W}} [\alpha \cdot P(W, T) - \beta \cdot C(W, T)] \tag{1}$$

where $P(W, T)$ represents the agentic workflow $W$'s performance on task $T$ (e.g., accuracy), $C(W, T)$ denotes the operational cost (inference cost or token consumption), and $\alpha, \beta$ are balancing hyperparameters that reflect the relative importance of performance and cost.

We approach this as a search problem within a search space that includes LLM agents (LLMs with specific system prompts and tools) and their communication patterns (such as sequential execution, conditional execution, and loops). This creates a discrete and potentially infinite search space. The most common way to address this is using human expert priors to design specific workflows, e.g., Camel (Li et al., 2023), OAgent (Zhu et al., 2025), ReAct (Yao et al., 2023), LLM debate (Du et al., 2023), etc. For automatic methods, there have been methods to automatically optimize the prompt (Khattab et al., 2024) and also automatically design the workflow (Zhang et al., 2025e;b).

Following AFlow (Zhang et al., 2025e), we adopt Monte Carlo Tree Search (MCTS) to search this discrete search space. The entire search process follows these steps (illustrated in Figure 2):

---
**Algorithm 1** OneFlow (MCTS with Dual Meta-LLMs)
---
**Require:** Task $T$, validation set $D_v$, iterations $K$, weights $\alpha, \beta$
**Ensure:** Best workflow $W^*$
1: $W_{\text{IO}} \leftarrow$ single-LLM input-output; Eval($W_{\text{IO}}$ on $D_v$); init tree with $W_{\text{IO}}$
2: **for** $k = 1$ to $K$ **do**
3:     $\mathcal{C} \leftarrow \{W_{\text{IO}}\} \cup$ Top-$(n-1)$ by score $S(W) = \alpha P - \beta C$
4:     $W \leftarrow$ Select($\mathcal{C}$)                       ▷ Selection; App. A.2
5:     $W_d \leftarrow$ Designer($W$, errors($W$))         ▷ Meta-LLM 1; App. A.3
6:     $W_{new} \leftarrow$ Reviewer($W_d, \mathcal{C}$)                 ▷ Meta-LLM 2
7:     Eval($W_{new}$ on $D_v$); attach $W_{new}$; backpropagate
8: **end for**
9: **return** $\arg\max_W \alpha P(W, T) - \beta C(W, T)$ over explored nodes
---

*Initialization.* We begin with the simplest possible workflow: $W_{\text{Input-Output}}$, which directly uses a single agent to answer questions in a straightforward input-output manner. We evaluate this workflow on a validation dataset $D_v$ (where $|D_v|$ means 20% samples of a specific dataset, e.g., the HumanEval dataset has 164 questions, so the validation dataset has 33 questions) to measure its performance, including both performance metrics (such as accuracy) and cost metrics (such as inference cost in USD). This input-output workflow becomes the root node in our Monte Carlo tree (workflow 1 in Figure 2), storing both performance metrics and examples of failure cases.

*Iterative Monte Carlo Tree Search (MCTS) Process.* We then follow a standard four-stage MCTS approach, adapted for workflow optimization. *1. Selection.* Choose a workflow $W$ from the tree based on a performance-based probability distribution. *2. Expansion.* Based on the selected workflow $W$, generate a new workflow $W_{new}$ using dual meta-LLMs that collaboratively design improved workflows. *3. Evaluation.* Test the new workflow $W_{new}$ on validation data to obtain performance and cost metrics. *4. Backpropagation.* Compare the performance and cost metrics of the new workflow $W_{new}$ with the parent workflow $W$, and record this modification to the parent workflow $W$ to avoid redundant designs. This iterative process continues until reaching a maximum number of iterations (we set it to 20 iterations in our experiments), producing a diverse set of automatically generated workflows tailored to the specific task.

*Dual Meta-LLM Architecture.* As illustrated in Figure 2, OneFlow employs two specialized meta-LLMs during the expansion phase: a Creative Designer that proposes performance-focused workflow improvements, and a Critical Reviewer that refines these proposals to optimize cost-efficiency. This collaborative design balances the performance-cost trade-off in Equation 1. Details about algorithm can be found in the appendix MCTS optimization (Section A.2) and dual meta-LLMs for balanced performance and cost optimization (Section A.3).

**OneFlow includes two stages: first, it searches for the optimized workflow, then performs single LLM implementation of the optimized workflow.** We use OneFlow and OneFlow (single-agent execution) to clearly distinguish these two phases.

## 4 EXPERIMENTS

### 4.1 EXPERIMENTAL SETUP

**Datasets.** We evaluate our approach across a diverse set of benchmarks spanning multiple domains to assess generalization capabilities. Our evaluation suite includes: (*i*) *code generation tasks*: MBPP and HUMANEVAL; (*ii*) *mathematical reasoning*: GSM8K and MATH; (*iii*) *question answering*: HOTPOTQA and DROP; (*iv*) *domain-specific reasoning*: SHOPPING-MMLU; and (*v*) *real-world planning and tool use*: TRAVELPLANNER.

**Evaluation Metrics.** We assess both task performance and computational efficiency. For code generation, we report `pass@1` accuracy; for general question answering, we use F1 score; for mathematical tasks, we report solve rate (%); for Shopping-MMLU, we use accuracy. For TravelPlanner, we use task success rate (%). Computational cost is measured as USD token expenditure per work-

flow, accounting for both input and output token usage. To ensure statistical reliability, we conduct three independent trials and report mean values with standard deviations.

**Baseline Methods.** We compare against four categories of approaches: (1) *Manual baselines*: Direct input-output (IO) prompting, chain-of-thought (CoT) prompting (Wei et al., 2022), self-consistency (CoT) and MultiPersona (Wang et al., 2024b); (2) *Automated multi-agent frameworks*: AFlow (Zhang et al., 2025e) and our proposed OneFlow; (3) *Heterogeneous multi-agent systems*: AFlow-optimized workflows using GPT-4o-mini and Claude-3.5-Haiku as heterogeneous executors; (4) *Single-LLM implementations*: Following Section 3.2, we execute multi-agent workflows (AFlow and OneFlow) using a single LLM agent.

**Model Configuration.** Following established practices (Zhang et al., 2025e), we use GPT-4o-mini as the primary executor LLM across all methods with temperature set to 0. For robustness validation, we additionally evaluate with Claude-3.5-Haiku (results are in the appendix) and Qwen-3 8B (to verify findings on open-weight models). Methods requiring workflow optimization (e.g., AFlow) employ Claude-4.0-Sonnet as the designer/optimizer with 20 optimization rounds.

**Heterogeneous Multi-agent Workflow Implementation.** To investigate model heterogeneity effects, we leverage AFlow (Zhang et al., 2025e) to automatically design heterogeneous multi-agent workflows. For homogeneous multi-agent workflows, all the executor LLMs are the same (either gpt-4o-mini or claude 3.5 haiku). In this heterogeneous setting, the workflow has two executor models at the same time, GPT-4o-mini and Claude-3.5-Haiku. Claude-4.0-Sonnet serves as the workflow designer. Optimization iterations are capped at 20 rounds with temperature 0. API interactions utilize OpenAI Chat Completions and Anthropic interfaces. System prompts for the optimizing process are detailed in Appendix A.6.

**KV Cache and Cost Estimation for Single-agent Implemented Multi-agent Workflows.** Implementing KV-cache optimization (Section 3.2) typically requires open-weight LLMs. Since we employ closed-weight LLMs (GPT-4o-mini) via API calls, we simulate ideal KV-cache costs by utilizing the final conversation states (the final message list). Cost calculations follow OpenAI's official tokenization for GPT-4o-mini, categorizing user prompts as input tokens and assistant responses as output tokens to estimate the theoretical KV-cache cost. For open-weight models (Qwen-3 8B), we explicitly measure latency and throughput using vLLM with KV cache enabled.

## 4.2 EXPERIMENTAL RESULTS AND ANALYSIS

Table 1: Main results on public benchmarks with GPT-4o mini as executors. Values are mean $\pm$ std over three runs, in percentage (0–100). Code tasks report pass@1; QA tasks report F1; solve rate (%) for math. Best per column in **bold**; runner-up per column underscored.

| Method | CODE | | MATH | | QA | |
|---|---|---|---|---|---|---|
| | HumanEval | MBPP | GSM8K | MATH | HotpotQA | DROP |
| *Manual baselines* | | | | | | |
| IO | 89.1 $\pm$ 0.4 | 72.6 $\pm$ 0.3 | 87.0 $\pm$ 0.1 | 51.3 $\pm$ 0.6 | 71.1 $\pm$ 0.5 | 66.2 $\pm$ 0.6 |
| CoT (Wei et al., 2022) | 90.3 $\pm$ 1.2 | 73.3 $\pm$ 0.0 | 87.1 $\pm$ 0.2 | 50.9 $\pm$ 0.3 | 71.2 $\pm$ 1.0 | 78.9 $\pm$ 0.4 |
| CoT SC (5-shot) | 89.8 $\pm$ 0.9 | 71.9 $\pm$ 1.0 | 92.6 $\pm$ 0.8 | 37.7 $\pm$ 1.2 | 67.3 $\pm$ 0.3 | 79.4 $\pm$ 0.1 |
| MultiPersona (Wang et al., 2024b) | 89.1 $\pm$ 0.2 | 73.3 $\pm$ 0.3 | 87.1 $\pm$ 0.1 | 50.9 $\pm$ 0.3 | 71.2 $\pm$ 1.0 | 78.9 $\pm$ 0.4 |
| *Automatically designed multi-agent frameworks* | | | | | | |
| AFlow (Zhang et al., 2025e) | 90.1 $\pm$ 0.0 | 78.8 $\pm$ 0.7 | **93.6 $\pm$ 0.5** | **55.6 $\pm$ 0.3** | 72.1 $\pm$ 0.2 | **83.1 $\pm$ 0.3** |
| OneFlow | 91.6 $\pm$ 0.8 | 81.1 $\pm$ 0.4 | 93.0 $\pm$ 0.4 | 53.4 $\pm$ 1.4 | **73.5 $\pm$ 0.5** | 81.1 $\pm$ 0.8 |
| *Single-LLM implementation of multi-agent workflow* | | | | | | |
| AFlow (single-agent execution) | 91.1 $\pm$ 1.6 | 78.8 $\pm$ 0.7 | 92.9 $\pm$ 0.1 | 53.8 $\pm$ 0.9 | 68.4 $\pm$ 0.1 | 81.1 $\pm$ 0.7 |
| OneFlow (single-agent execution) | **92.1 $\pm$ 0.4** | **81.4 $\pm$ 0.6** | 93.3 $\pm$ 0.1 | 54.1 $\pm$ 0.7 | **73.5 $\pm$ 0.5** | 81.7 $\pm$ 0.7 |

### 4.2.1 PERFORMANCE ON PUBLIC BENCHMARKS

**A single-agent implementation of a multi-agent workflow can match multi-agent performance.** Table 1 summarizes results with GPT-4o mini (pass@1 for code; F1 for QA; solve rate (%) for math). Across the board, automatically designed multi-agent workflows and their single-agent executions substantially outperform manual baselines, highlighting the value of automated design. Notably, executing AFlow- and OneFlow-designed workflows with *a single agent* matches or slightly exceeds

their multi-agent counterparts, consistent with the hypothesis in Section 3.1: in homogeneous settings, a single model can faithfully simulate agent roles via multi-turn conversations. Our proposed OneFlow, typically when using the single-agent execution setting, shows superior performance compared to all the homogeneous workflow baselines.

### 4.2.2 COST ON PUBLIC BENCHMARKS

Table 2: Inference cost (USD) on public benchmarks with GPT-4o mini. Values are mean $\pm$ std over three runs (lower is better). Best per column in **bold**; runner-up per column underscored.

| Method | CODE | | MATH | | QA | |
|---|---|---|---|---|---|---|
| | HumanEval | MBPP | GSM8K | MATH | HotpotQA | DROP |
| *Manual baselines* | | | | | | |
| CoT SC (5-shot) | $0.103 \pm 0.000 | $0.177 \pm 0.000 | $1.265 \pm 0.001 | $1.561 \pm 0.009 | $1.201 \pm 0.001 | $0.613 \pm 0.001 |
| MultiPersona | $0.099 \pm 0.000 | $0.226 \pm 0.001 | $\underline{0.429 \pm 0.005} | **$0.330 $\pm$ 0.021** | $0.415 \pm 0.000 | $\underline{0.301 \pm 0.000} |
| *Automatically designed multi-agent frameworks* | | | | | | |
| AFlow | $0.198 \pm 0.003 | $0.393 \pm 0.002 | $1.134 \pm 0.001 | $2.343 \pm 0.036 | $1.438 \pm 0.000 | $0.771 \pm 0.001 |
| OneFlow | $\underline{0.026 \pm 0.000} | $\underline{0.070 \pm 0.005} | $0.623 \pm 0.001 | $0.819 \pm 0.007 | **$0.278 $\pm$ 0.000** | $0.322 \pm 0.000 |
| *Single-LLM implementation of multi-agent workflow* | | | | | | |
| AFlow (single-agent execution) | $0.198 \pm 0.004 | $0.283 \pm 0.001 | $0.697 \pm 0.001 | $2.039 \pm 0.028 | $0.530 \pm 0.001 | $0.345 \pm 0.001 |
| OneFlow (single-agent execution) | **$0.020 $\pm$ 0.000** | **$0.063 $\pm$ 0.004** | **$0.387 $\pm$ 0.000** | $\underline{0.677 \pm 0.002} | **$0.278 $\pm$ 0.000** | **$0.284 $\pm$ 0.001** |

**Single-agent execution is substantially more efficient and cheaper than multi-agent execution.** Table 2 shows that single-LLM execution dramatically reduces cost at comparable performance (Table 1), largely due to KV-cache reuse across agent turns in homogeneous workflows. Without single-LLM execution, OneFlow is more cost-efficient than AFlow; when executed as a single LLM, both AFlow and OneFlow realize cost gains and still maintain performance. For OneFlow, the performance even slightly increases with single-agent execution, thanks to KV sharing, which provides more context to the agent and generates better results. Table 10 further breaks down input/output tokens and explains where the savings arise.

Table 3: Executor-specific results and heterogeneous baseline on public benchmarks. Values are mean $\pm$ std over three runs, in percentage (0–100). Best per column in **bold**; best per base model type per column underscored.

| Method (Executor) | CODE | | MATH | | QA | |
|---|---|---|---|---|---|---|
| | HumanEval | MBPP | GSM8K | MATH | HotpotQA | DROP |
| *GPT-4o mini-based* | | | | | | |
| AFlow (GPT-4o mini) | 90.1 $\pm$ 0.0 | 78.8 $\pm$ 0.7 | **93.6 $\pm$ 0.5** | $\underline{55.6 \pm 0.3}$ | 72.1 $\pm$ 0.2 | $\underline{83.1 \pm 0.3}$ |
| OneFlow (GPT-4o mini) | $\underline{91.6 \pm 0.8}$ | 81.1 $\pm$ 0.4 | 93.0 $\pm$ 0.4 | 53.4 $\pm$ 1.4 | $\underline{73.5 \pm 0.5}$ | 81.1 $\pm$ 0.8 |
| AFlow (GPT-4o mini, single-agent) | 91.1 $\pm$ 1.6 | 78.8 $\pm$ 0.7 | 92.9 $\pm$ 0.1 | 53.8 $\pm$ 0.9 | 68.4 $\pm$ 0.1 | 81.1 $\pm$ 0.7 |
| OneFlow (GPT-4o mini, single-agent) | **92.1 $\pm$ 0.4** | $\underline{81.4 \pm 0.6}$ | 93.3 $\pm$ 0.1 | 54.1 $\pm$ 0.7 | $\underline{73.5 \pm 0.5}$ | 81.7 $\pm$ 0.7 |
| *Claude 3.5 Haiku-based* | | | | | | |
| AFlow (Claude 3.5 Haiku) | 90.8 $\pm$ 0.0 | 83.6 $\pm$ 0.0 | 91.2 $\pm$ 0.0 | 50.5 $\pm$ 0.0 | 74.6 $\pm$ 0.0 | 86.8 $\pm$ 0.0 |
| OneFlow (Claude 3.5 Haiku) | $\underline{91.6 \pm 0.0}$ | **84.4 $\pm$ 0.0** | $\underline{93.0 \pm 0.0}$ | $\underline{51.3 \pm 0.0}$ | $\underline{74.7 \pm 0.0}$ | **87.5 $\pm$ 0.0** |
| *Heterogeneous baseline* | | | | | | |
| AFlow (Heterogeneous: GPT-4o mini + Claude 3.5 Haiku) | 87.0 $\pm$ 0.8 | 80.0 $\pm$ 0.3 | **93.6 $\pm$ 0.3** | **55.7 $\pm$ 0.6** | **75.1 $\pm$ 0.5** | 85.5 $\pm$ 0.5 |

### 4.2.3 PILOT STUDY ON AUTOMATICALLY DESIGNED HETEROGENEOUS MULTI-AGENT WORKFLOWS

**Performance is largely bounded by the best homogeneous workflow.** We conduct a pilot study using an automatically designed heterogeneous multi-agent workflow with GPT-4o mini and Claude 3.5 Haiku collaboratively working within the workflow. We notice that the performance of this heterogeneous workflow is largely bounded by the homogeneous multi-agent workflow. E.g., the performance on DROP achieves an F1-score of 85.5, even outperforming all GPT-4o mini-based methods (with AFlow utilizing GPT-4o mini having an F1 of 83.1), but is still bounded by the best performance of the Claude 3.5 Haiku-based homogeneous workflow (specifically OneFlow, with an F1 of 87.5). However, we want to emphasize that this pilot study uses automatically generated heterogeneous multi-agent workflows, which are not perfectly optimized. When deploying multi-agent systems in the real world, **well-designed** heterogeneous multi-agent workflows can be very beneficial. For example, for simple tasks, we can use a small LLM agent to handle them, while for complex tasks, we may route them to a strong reasoning model. ***Implications.*** In homogeneous

settings, single-agent execution is a strong, cost-efficient baseline and even matched the performance of AFlow-optimized heterogeneous workflows. Two directions stand out: (1) train a single agent to execute complex workflows end-to-end; (2) design effective heterogeneous workflows that mix models of different strengths and costs to collaborate efficiently despite no KV sharing.

### 4.2.4 KV CACHE REUSE WITH OPEN-WEIGHT LLMS

Table 4: Results on HumanEval with Qwen-3 8B. Single-agent execution maintains performance and efficiency thanks to KV cache reuse. Scores are averaged over three independent runs.

| Method | pass@1 | Avg Latency (s) | Throughput (samples/s) | Avg Input Tokens | Avg Output Tokens |
|---|---|---|---|---|---|
| CoT | 83.5% | 2.60 | 7.66 | 213 | 74 |
| CoT SC × 5 | 83.7% | 17.44 | 1.32 | 1748 | 502 |
| MultiPersona | 84.7% | 32.38 | 0.75 | 1722 | 846 |
| AFlow (multiple stateless api calls) | 86.8% | 54.98 | 0.17 | 2302 | 1753 |
| AFlow (Single-agent execution) | 90.5% | 53.53 | 0.18 | 3269 | 1739 |
| OneFlow (multiple stateless api calls) | 87.0% | 4.31 | 2.47 | 920 | 148 |
| OneFlow (Single-agent execution) | 87.4% | 4.83 | 1.70 | 1288 | 159 |

To validate our findings beyond proprietary models, we conduct experiments using the open-weight Qwen-3 8B model with vLLM, setting the context window to 16k to reflect typical resource constraints. As shown in Table 4, single-agent execution of both AFlow and OneFlow maintains or improves performance (pass@1 on HumanEval) compared to multiple stateless API calls. Crucially, while multi-turn conversations increase the average input tokens due to history accumulation (+967 for AFlow, +368 for OneFlow), the inference efficiency (latency and throughput) remains stable thanks to KV cache reuse. We note that Qwen-3 8B tends to generate longer responses in multi-turn settings compared to single-turn, slightly offsetting efficiency gains seen with GPT-4o-mini, but the core benefit of KV cache sharing in homogeneous workflows is confirmed.

### 4.2.5 ADDITIONAL EXPERIMENTS ON THE TOOL-USING BENCHMARK: TRAVELPLANNER

We additionally evaluate on TravelPlanner (Xie et al., 2024), a context- and tool-intensive benchmark for real-world planning, to assess whether single-agent execution of homogeneous multi-agent workflows remains competitive. A single LLM executing the AFlow and OneFlow workflows matches the task success rate of their original multi-agent counterparts while incurring lower inference cost. The workflow uncovered by OneFlow is especially compact and efficient, further reducing cost. The results are shown in Figure 3.

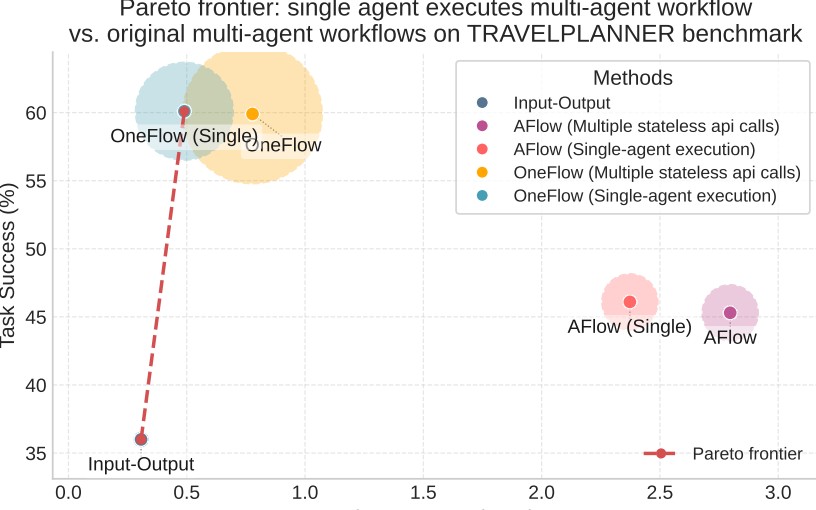

Figure 3: Pareto frontier: single agent executes multi-agent homogeneous workflow vs. original multi-agent workflows on TravelPlanner (Xie et al., 2024) benchmark. All the workflows are searched by Claude 4 Sonnet. All the workflows are executed by GPT-4o mini.

## 5 RELATED WORK

**Multi-Agent Workflows and Task Decomposition.** Multi-agent systems organize multiple LLM agents with complementary roles and communication patterns to solve complex tasks more reliably than a single-pass input-output baseline. Foundational designs include role-playing and tool-augmented collaboration (e.g., CAMEL, OAgent, ReAct, debate) that decompose tasks into specialized stages and coordinate agents via sequential, iterative, or deliberative interactions (Li et al., 2023; Zhu et al., 2025; Yao et al., 2023; Du et al., 2023). A key empirical trend is that *most* frameworks are **homogeneous**: agents share the same base LLM and differ by prompts, tools, and routing logic (Zhuge et al., 2024; Liu et al., 2024). Heterogeneous systems, where agents use different base models, have been explored to leverage model diversity (Ye et al., 2025a; Zhang et al., 2025a; Wang et al., 2024a; Jiang et al., 2023; Chen et al., 2023), but many such approaches emphasize ensembling or discussion rather than end-to-end **execution** with explicit control flow. This paper examines when a single agent can faithfully simulate homogeneous workflows via multi-turn conversations, and clarifies where heterogeneity may bring benefits and costs.

**Automatic Design of Multi-Agent Workflows.** Reducing human effort in workflow construction has led to three complementary directions: (1) *prompt optimization*, (2) *communication/topology optimization*, and (3) *automatically workflow search/generation*. For prompt optimization, DSPy formalizes modular prompt programs with compilation-time optimization, and TextGrad proposes gradient-inspired improvements (Khattab et al., 2024; Yuksekgonul et al., 2025). For communication/topology optimization, GPTSwarm explores graph-structured agent teams with iterative refinement; Dylan performs dynamic agent selection; Agent-Prune prunes redundant edges; and G-Designer learns task-adaptive topologies (Zhuge et al., 2024; Liu et al., 2024; Zhang et al., 2025c;d). For automatic workflow design, ADAS is the first to propose this idea and conducts heuristic search; AFlow uses MCTS with named nodes; MaAS optimizes a distribution over architectures via a controller; AgentSquare extends this paradigm; and recent methods generate workflows directly via fine-tuning or continuous optimization (MAS-GPT, Flow-reasoner, ScoreFlow), or train weaker meta-agents to design for stronger executors (Weak-for-strong) (Hu et al., 2025; Zhang et al., 2025e;b; Shang et al., 2025; Ye et al., 2025b; Gao et al., 2025; Wang et al., 2025b; Nie et al., 2025). While these systems substantially reduce manual design effort, the vast majority assume homogeneous executors; some incorporate heterogeneous options only tangentially (e.g., MAS-Router, limited heterogeneity settings in AFlow) (Yue et al., 2025; Zhang et al., 2025e). Our work complements this line by showing that a strong single-LLM simulator provides a competitive, cost-efficient baseline for many automatically designed *homogeneous* workflows, while clarifying when heterogeneity remains necessary. We also note an orthogonal trend toward smaller, cost-efficient models that further motivates cost-aware workflow design (Belcak et al., 2025).

**KV Cache and Single-LLM Execution.** Transformer KV caching reuses previously computed key/value states to avoid repeated prefill, enabling substantial speedups in autoregressive decoding. When multiple agents share the *same* base model (homogeneous workflows), a single-LLM execution can maintain one conversation with shared KV cache, avoiding redundant re-encoding across agent turns and often improving consistency. In contrast, *heterogeneous* workflows cannot share KV states across different base models, limiting these gains and complicating end-to-end training. Prior work has analyzed caching behaviors and memory allocation (e.g., attention sinks) and proposed structured caches for long or graph-structured contexts (Xiao et al., 2024; Wang et al., 2025a). These observations motivate our study: if behavior can be preserved (under mild conditions) and caches can be shared, single-LLM simulations offer an informative and strong baseline for homogeneous agentic workflows.

## 6 CONCLUSION

In homogeneous multi-agent workflows, a single LLM can role-play agents via multi-turn conversations, reuse a shared KV cache, and match or slightly exceed multi-agent performance at substantially lower cost. **OneFlow** helps discover compact workflows and, when executed by a single LLM, yields additional efficiency gains. While single-LLM simulation cannot realize true heterogeneity, our pilot shows it can even match the performance of AFlow-optimized heterogeneous workflows; we view (1) **training single agents for end-to-end execution** and (2) **principled heterogeneous composition** as complementary, promising directions in LLM multi-agent system research.

ETHICS STATEMENT

This research primarily focuses on evaluating the effectiveness of using a single LLM (large language model) agent to perform multi-agent workflows, with an emphasis on empirical validation. It relies solely on existing LLMs and does not involve training, fine-tuning model weights, or creating new LLMs. As such, the work does not raise any novel ethical considerations or societal impacts beyond those already well documented in relation to large-scale language models more broadly.

REPRODUCIBILITY STATEMENT

We provide pseudocode and a detailed explanation of OneFlow in the main methodology section, along with a visual illustration for clarity. We also include the exact prompts used, as well as the model settings for both the executor model and the optimization model.

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

# A APPENDIX

## A.1 THE USE OF LARGE LANGUAGE MODELS

Large language models were used solely for sentence-level proofreading. All research ideation and paper writing were conducted entirely by the authors.

## A.2 MONTE CARLO TREE SEARCH FOR WORKFLOW OPTIMIZATION

As mentioned above and Figure 2, we employ MCTS to systematically explore the space of possible workflows, treating each workflow configuration as a node in the search tree. Specifically:

**1. Selection:** At each iteration, we form a candidate set consisting of the initial workflow $W_{\text{Input-Output}}$ together with the top 3 best-performing workflows from the Monte Carlo Tree. Retaining the initial workflow $W_{\text{Input-Output}}$ in the candidate set helps the framework maintain exploration of the workflow space. We then select one Workflow $W$ workflow from the candidate set for expansion. Following AFlow (Zhang et al., 2025e), we employ a mixed-probability selection strategy that combines uniform distribution (for exploration) and score-based weighting (for exploitation):

$$p_i = \lambda \cdot \frac{1}{n} + (1 - \lambda) \cdot \frac{\exp(\alpha \cdot s_i)}{\sum_j \exp(\alpha \cdot s_j)} \tag{2}$$

where $s_i$ is the score of workflow $i$, $\alpha$ controls the sharpness of the distribution, $\lambda$ balances exploration and exploitation, and $n \leq 4$ represents the number of candidates. This formula ensures that better-performing workflows have a higher probability of being selected as Workflow $W$.

**2. Expansion:** This phase leverages the dual meta-LLM architecture (detailed in Appendix A.3). At each expansion, the designer meta-LLM proposes a new workflow design $W_{new1}$ (represented in Python code with detailed comments) based on the selected workflow $W$ from the previous selection step and the failed cases in the validation dataset $D_v$. Next, the critic meta-LLM reviews the proposed workflow design $W_{new1}$ to assess its validity and efficiency by examining the code comments and Python implementation. The critic also compares the proposal $W_{new1}$ with candidate workflows from the selection stage. Finally, the critic meta-LLM proposes an improved workflow $W_{new2}$ based on the designer's proposal $W_{new1}$, writing detailed comments in the workflow code to record changes and their rationale. The output of this stage is a runnable workflow.py file that implemented $W_{new2}$.

**3. Evaluation.** The proposed workflow $W_{new2}$ is then evaluated on the validation dataset $D_v$ to obtain performance metrics (e.g., accuracy), cost metrics (e.g., token consumption), and failed samples (including the specific reasoning processes of the failures).

**4. Backpropagation:** The performance and cost, and failure cases are stored in the current workflow node $W_{new2}$. The relative success or failure compared to the parent workflow node $W$ is stored in the parent node $W$ (we call this backpropagation) to avoid proposing similar workflows. The optimization process continues until a maximum number of iterations is reached (20).

### A.3 DUAL META-LLMS FOR BALANCED PERFORMANCE AND COST OPTIMIZATION

We design two specialized meta-LLMs that collaboratively design new workflows (the right panel of Figure 2 shows how this dual meta-LLM framework works):

**Meta-LLM 1: Creative Designer.** This meta-LLM's goal is to creatively, even disruptively, improve a workflow. It receives the code and prompt of the current agentic workflow $W_{\text{current}}$, the workflow's performance score (e.g., accuracy), any previous modifications based on $W_{\text{current}}$, the corresponding performance score, and a sample of incorrectly answered questions, $D_{\text{error}}$, including the questions, the reasoning process when the workflow failed to answer the question, and the ground truth answer. It also receives instructions on how to write a runnable `workflow.py` file, how to define system prompts for LLM agents, and how to utilize existing operators (e.g., chain-of-thought reasoning, ensembling, executing Python code, etc.). With this information, the Creative Designer is prompted to make creative modifications to improve workflow performance and add comments to the workflow code explaining the rationale for the changes.

$$W_{\text{creative designer}} = \text{Designer}(W_{\text{current}}, D_{\text{error}}, \text{instructions}) \quad (3)$$

The creative designer's setting is similar to the AFlow (Zhang et al., 2025e) setting, with two modifications: 1) the creative designer is prompted to write code comments explaining the rationale for modifications, which can inform the other meta-LLM (the critical reviewer), and 2) the creative designer is given more detailed error logs, which include the specific reasoning process during the entire workflow's execution. This helps the creative designer better understand the workflow's failure cases. Empirically, we found that the creative designer alone can propose workflows with excellent performance, but the inference cost of the workflow is high.

**Meta-LLM 2: Critical Reviewer.** The Critical Reviewer's goal is to review and critique the creative designer's proposed workflow and modify it to avoid mistakes and improve cost-efficiency. To accomplish this, we provide the critical reviewer with the same information as the creative designer. Additionally, we give the critical reviewer the creative designer's proposed workflow (a runnable `workflow.py` file, including code and comments) and the cost and performance metrics of existing workflow candidates (the top 3 best-performing workflows from the Monte Carlo tree, plus the initial workflow $W_{\text{Input-Output}}$). We then prompt the critical reviewer to carefully examine the creative designer's proposed workflow, using the existing workflow candidates' cost and performance as reference points. The critical reviewer proposes a new workflow design based on the designer's proposal, writing detailed comments in the workflow code to document changes and their rationale. The goal is to avoid mistakes and improve cost-efficiency. The output of this stage is a runnable `workflow.py` file. The critical reviewer also writes reflections on the improved workflow. Since the critical reviewer is given more global information and cost data, it can provide a broader view of the workflow's performance and cost trade-offs to guide future rounds of workflow design.

$$W_{\text{critical reviewer}} = \text{Reviewer}(W_{\text{creative designer}}, W_{\text{candidates}}, \text{cost\_metrics}) \quad (4)$$

We use the critical reviewer's output workflow as the final workflow of this round.

### A.4 RESULTS ON CLAUDE 3.5 HAIKU

To further evaluate the effectiveness of our proposed OneFlow framework, we conduct experiments on Claude 3.5 Haiku. The results are shown in Table 5 and Table 6.

**Performance on Shopping-Specific Tasks.** Table 7 demonstrates that our OneFlow approach achieves state-of-the-art performance across 9 out of 10 shopping-specific tasks, outperforming both single-LLM baselines and existing multi-agent frameworks. Notably, we observe substantial improvements over the strongest baseline (AFlow) in critical shopping domains: Product Selection (+4.0% absolute improvement, 0.678 vs. 0.638), Sentiment Analysis (+2.5%, 0.777 vs. 0.752), and Multilingual Query Understanding (+4.8%, 0.569 vs. 0.521). These gains are particularly significant given the complexity of shopping-specific reasoning tasks, where domain knowledge and nuanced understanding of user intent are crucial.

Table 5: Results on Claude 3.5 Haiku. Performance comparison on public benchmarks. We report pass@1 accuracy for code generation tasks (HumanEval, MBPP) and F1 scores for question-answering tasks (HotpotQA, DROP), solve rate (%) for GSM8K, MATH. OneFlow achieves competitive or superior performance across most benchmarks. Results are averaged over three independent runs.

| Method | HumanEval | MBPP | GSM8K | MATH | HotpotQA | DROP |
|---|---|---|---|---|---|---|
| IO | 0.870 | 0.745 | 0.875 | 0.329 | 0.635 | 0.640 |
| AFlow (Zhang et al., 2025e) | 0.908 | 0.836 | 0.912 | 0.505 | 0.746 | 0.868 |
| **OneFlow (Ours)** | **0.916** | **0.844** | **0.930** | **0.513** | **0.747** | **0.875** |

Table 6: Results on Claude 3.5 Haiku. Cost comparison on public benchmarks. Values represent total inference cost in USD (lower is better). OneFlow achieves substantial cost reductions while maintaining competitive performance. Results are averaged over three independent runs.

| Method | HumanEval | MBPP | GSM8K | MATH | HotpotQA | DROP |
|---|---|---|---|---|---|---|
| AFlow (Zhang et al., 2025e) | **0.20** | 2.39 | 6.68 | **3.55** | 6.55 | 3.46 |
| OneFlow (Ours) | 0.25 | **0.58** | **5.83** | 4.33 | **1.47** | **3.14** |

Table 7: Performance comparison on Shopping-MMLU tasks using accuracy as the evaluation metric. OneFlow achieves state-of-the-art performance across 9 out of 10 shopping-specific tasks. Results are averaged over three independent runs. Task abbreviations: Product Sel. (applicable product selection), Sentiment (aspect-based sentiment classification), Attribute (implicit attribute selection), Multi-lang (multilingual query product semantic understanding), Prod. Comp. (product complements), Query (query product semantic classification), Brand (related brands selection), Keyword (related keyword intent selection), Rev. Help (review helpfulness selection), Rev. Sent. (reviews overall sentiment selection). Best results are shown in **bold**.

| Method | Product Sel. | Sentiment | Attribute | Multi-lang | Prod. Comp. | Query | Brand | Keyword | Rev. Help | Rev. Sent. |
|---|---|---|---|---|---|---|---|---|---|---|
| IO | 0.638 | 0.668 | 0.650 | 0.465 | 0.713 | 0.464 | 0.681 | 0.728 | 0.586 | 0.578 |
| CoT (Wei et al., 2022) | 0.647 | 0.712 | 0.700 | 0.441 | 0.700 | 0.455 | **0.761** | 0.713 | 0.586 | 0.566 |
| SC(CoT X 5) (Wang et al., 2023) | 0.656 | 0.731 | 0.713 | 0.473 | 0.703 | 0.491 | 0.761 | 0.725 | 0.598 | 0.566 |
| AFlow (Zhang et al., 2025e) | 0.638 | 0.752 | 0.769 | 0.521 | 0.736 | 0.530 | **0.761** | 0.713 | 0.550 | 0.698 |
| OneFlow (Ours) | **0.678** | **0.777** | **0.775** | **0.569** | **0.756** | **0.567** | 0.756 | **0.783** | **0.609** | **0.718** |

Table 8: Cost efficiency comparison on Shopping-MMLU tasks measured in USD inference cost (lower is better). Results are averaged over three independent runs. OneFlow demonstrates superior cost efficiency across all shopping-specific tasks while maintaining competitive performance. Best results are shown in **bold**.

| Method | Product Sel. | Sentiment | Attribute | Multi-lang | Prod. Comp. | Query | Brand | Keyword | Rev. Help | Rev. Sent. |
|---|---|---|---|---|---|---|---|---|---|---|
| SC(CoT X 5) (Wang et al., 2023) | 1.87 | 1.78 | 1.62 | 1.84 | 1.89 | 1.55 | 1.34 | 2.00 | 1.53 | 2.78 |
| AFlow (Zhang et al., 2025e) | 1.47 | 1.83 | 2.20 | 1.32 | 1.39 | 0.63 | **0.23** | **0.39** | 0.86 | 2.75 |
| OneFlow (Ours) | **0.26** | **0.48** | **0.63** | **0.46** | **0.42** | **0.47** | 0.32 | 0.60 | **0.31** | **0.74** |

## A.5 SYSTEM PROMPT FOR ONEFLOW

### A.5.1 SYSTEM PROMPT FOR THE CREATIVE DESIGNER META-LLM

The following designer prompts are adapted from AFlow (Zhang et al., 2025e).

```
WORKFLOW_OPTIMIZE_PROMPT_DESIGNER = """You are building a workflow
    Graph (nodes are llm agents, edges are the flow of information)
    and corresponding Prompt to jointly solve {type} problems.
Referring to the given graph and prompt, which forms an example of a
    {type} solution approach, please reconstruct and optimize them.
You can add, modify, or delete nodes, edges, or prompts. Include your
    single modification in XML tags in your reply. Ensure they are
    complete and correct to avoid runtime failures.
```

```
When optimizing, you can incorporate critical thinking methods such
    as review, revise, ensemble (generating multiple answers through
    different/similar prompts, then voting/integrating/checking the
    majority to obtain a final answer), brainstorming, expert-based
    reasoning, to solve the problem efficiently and effectively.
Consider Python's loops (for, while, list comprehensions),
    conditional statements (if-elif-else, ternary operators), and
    other programming techniques to enhance the workflow. Use logical
    and control flow (IF-ELSE, loops) for a more enhanced graphical
    representation.
You can design a workflow that adapts its approach based on the
    complexity of each problem. For example, using Python if-else
    statements to choose different solution strategies for easy vs.
    hard problems.

PROMPT REQUIREMENTS:
- Only generate prompts used by Custom operators in your graph
    (accessed as prompt_custom.YOUR_PROMPT_NAME)
- Built-in operators already have their own prompts – don't generate
    prompts for them
- Remove any unused prompts from prompt_custom
- Generated prompts must be complete with no placeholders

Output the modified graph and all necessary prompt_custom prompts.
It's crucial to include necessary context during the process.
Be creative and try to push the boundaries of the existing graph.
Write concise and informative inline comments for the code you
    modified to explain the logic and the purpose, why the
    modification is creative and effective. With all the comments you
    write, you should leave your name: Designer (modifying the round
    X graph).
If you encounter code comments written by other designers or critics,
    keep those that are useful for future reference.
"""

WORKFLOW_INPUT_DESIGNER = """
Here is a workflow graph and the corresponding prompt (prompt only
    related to the custom method) that has been tested on the
    previous round and its score (maximum score is 1). You must make
    further optimizations and improvements based on this graph. The
    modified graph must differ from the provided example, and the
    specific differences should be noted within the
    <modification>xxx</modification> section.\n
<sample>
    <experience>{experience}</experience>
    <modification>(such as:add /delete /modify/ ...)</modification>
    <score>{score}</score>
    <graph>{graph}</graph>
    <prompt>{prompt}</prompt>(contains prompts used by Custom
    operators)

    <operator_description>{operator_description}</operator_description>
</sample>

You are encouraged to use the operators provided, not limited to the
    custom operator.

In the graph, you should use the list process to record the
    intermediate output of the workflow. And return the process as
    the one of the output of the workflow.
Below are the logs of some results with the aforementioned workflow
    Graph encountered errors, which can be used as references for
    optimization:
```

```
{log}

In your design, you cannot hardcode the answer (e.g., remember the
    answer of a specific problem), which can lead to overfitting.
    Instead, provide principles rather than specific answers.
First, provide optimization ideas. **Only one detail point can be
    modified**, and no more than 5 lines of code may be changed per
    modification; you can not totally change the existing graph,
    extensive modifications are strictly prohibited to maintain
    project focus!
When introducing new functionalities in the graph, please make sure
    to import the necessary libraries or modules yourself, except for
    operator, prompt_custom, create_llm_instance, which have already
    been automatically imported.
No need to import the operator, prompt_custom, create_llm_instance,
    which have already been automatically imported.

**Under no circumstances should Graph output None for any field.**
Use self.custom methods to restrict your output format, rather than
    using code (outside of the code, the system will extract answers
    based on certain rules and score them).
It is very important to format the Graph output answers, you can
    refer to the standard answer format in the log.
Be creative and try your best to propose new ideas.

    """
```

## A.5.2 SYSTEM PROMPT FOR THE CRITICAL REVIEWER META-LLM

The following critic prompts are adapted from AFlow (Zhang et al., 2025e).

```
WORKFLOW_OPTIMIZE_PROMPT_CRITIC = """You are a great workflow critic
    and optimizer.
You are the analytical critic and practical optimizer. Focus on (1)
    the potential of the designer's idea to push the boundaries of
    the existing workflow solutions and (2) cost-effectiveness of the
    designer's proposed solution.
- The designer's idea has a great chance to help us push the
    boundaries of the existing graph and may find great solutions, so
    you should try to implement the designer's idea in a
    cost-efficient way, if necessary.
- Preserve the designer's innovative ideas while making them more
    cost-efficient, if necessary.
- You are encouraged to use the operators provided, not limited to
    the custom operator.
- Fully respect the designer's idea, especially when the designer
    proposed to use a predefined operators, you just use it and do
    not modify it. Some operators can be expensive, but if the
    designer proposed to use them, you should give them a try as it
    may have great potential of performance improvement.
- If a solution involves too many api calls (agent) e.g., > 5 api
    calls per problem and are not necessary, it can be very slow and
    costly, you can try to find a way to make it more cost-efficient.
    Overly expensive solutions are not allowed.
- You have access to the existing top performing workflow graphs and
    their performance and cost metrics. Consider the performance and
    cost of these existing graphs (they are the existing solutions
    for the same problem) when optimizing the designer's graph. You
    should not blindly reuse the existing top performing graphs, but
    use them as references to improve the cost-efficiency of the
    designer's idea, which may have great potential.

## Code Commentary Requirements:
```

```
- Add concise inline comments explaining your critical analysis and
    optimization rationale
- Include practical critiques at the code's end, drawing from
    top-performing graphs and best practices
- Preserve useful existing comments from other designers/critics,
    remove redundant ones
- Sign your comments as: "Critic (modifying round X workflow proposed
    by the designer)"

## Context and References:
When optimizing, refer to the designer's comments to understand the
    logic, purpose, and creative value of each modification. Consider
    how your optimization maintains the innovation while improving
    practical efficiency.

Referring to the designer's given graph and prompt for {type}
    problems, optimize them following the framework above."""

# Critic input prompts
WORKFLOW_INPUT_CRITIC = """
Here is a workflow graph and the corresponding prompt (prompt only
    related to the custom method) that has been proposed by a
    creative designer LLM (based on one existing workflow), which can
    be a great idea. You must respect and implement the designer's
    ideas. If the designer LLM's proposed workflow is already
    cost-effective, you just use it and do not change it. When the
    designer's idea is costly, you need to make further optimizations
    based on this graph to make it more cost-efficient. Try to
    preserve the original ideas as much as possible, but implement
    them in a great cost-efficient way if necessary. You are also
    provided with the existing top performing workflow graphs, their
    prompts, their performance metrics, most importantly, their cost
    metrics, on the same problems. Critically learn from their cost
    and strategy, and focus on improving the cost-efficiency of the
    designer's idea. Your improved workflow, by nature, is a great
    and cost-efficient implementation of the designer's idea.

{experience}

Only when you are very sure that the designer's idea is not good
    based on strong evidence, you can propose a new idea or
    fundamental changes. But in most cases, you should try to improve
    the designer's idea's cost-efficiency while preserving the
    original ideas, which can help us utilize the designer's
    innovative idea to push the boundaries of the existing workflow
    graph and find great solutions. The modified graph must differ
    from the provided graph (only exception is that, If the designer
    LLM's proposed workflow is already cost-effective, you just use
    it and do not change it.), and the specific differences should be
    noted within the <modification>xxx</modification> section, here
    is the workflow graph proposed by the designer LLM.
<sample>
    <modification>(such as:add /delete /modify/ ...)</modification>
    <graph>{graph}</graph>
    <prompt>{prompt}</prompt>(contains prompts used by Custom
    operators)

    <operator_description>{operator_description}</operator_description>
</sample>
```

```
In the graph, you should use the list process to record the
    intermediate output of the workflow. And return the process as
    the one of the output of the workflow.

You are encouraged to use the operators provided, not limited to the
    custom operator.

**ERROR LOGS FROM THE ORIGINAL WORKFLOW:**
The designer proposes an improved workflow (the graph above) based on
    an existing graph. The error logs of the old workflow that the
    designer is improving are:
{log}

**IMPORTANT:** Pay close attention to the error logs above. These
    logs show specific issues, failures, and problems encountered by
    the old workflow. Use these error logs to evaluate whether the
    designer's proposed improvements have potential to address these
    specific issues. In your design, you cannot hardcode the answer
    (e.g., remember the answer of a specific problem), which can lead
    to overfitting. Instead, provide principles rather than specific
    answers.

**CRITICAL FORMATTING RULE - READ THIS FIRST:**
NEVER combine formatting instructions with reasoning instructions in
    the same prompt. If you need both reasoning AND specific output
    formatting, you MUST use separate agents or code-based formatting.

**WHY:** LLMs cannot simultaneously reason deeply AND maintain strict
    output formats. When you ask an LLM to "think about this problem
    AND format your answer as X", it will prioritize thinking over
    formatting, causing format failures.

**SOLUTION:** Use one agent for reasoning, then use code (regex) or a
    separate formatting agent to ensure correct output format. This
    is non-negotiable for reliable results.

When introducing new functionalities in the graph, please make sure
    to import the necessary libraries or modules yourself, except for
    operator, prompt_custom, create_llm_instance, which have already
    been automatically imported.
No need to import the operator, prompt_custom, create_llm_instance,
    which have already been automatically imported.

**Under no circumstances should Graph output None for any field.**
Use self.custom methods to restrict your output format, rather than
    using code (outside of the code, the system will extract answers
    based on certain rules and score them).

Here are the existing top performing workflow graphs (the round 1 is
    not the top solution, it is a baseline solution) for the same
    problem and their performance and cost metrics. Do not blindly
    reuse the existing top performing graphs, but pay attention to
    their cost and performance, and use them as references to improve
    the cost-efficiency of the designer's idea if needed:
{top_solutions_context}
"""
```

## A.6 System prompt using for AFlow optimized heterogeneous multi-agent systems

The following guidance is adapted from AFlow (Zhang et al., 2025e).

```
# Optional guidance snippet appended when running in heterogeneous
    executor mode
HETERO_EXEC_GUIDANCE = """
\n[HETEROGENEOUS EXECUTOR MODE]
You can use two different executor LLMs within a single workflow:
    'claude-3-5-haiku-20241022' and 'gpt-4o-mini'.
- Define your Workflow __init__ to accept two configs (e.g.,
    llm_config_haiku35 and llm_config_4omini), create two Custom
    operators (self.custom_haiku35, self.custom_4omini), and choose
    between them per step based on problem characteristics.
- IMPORTANT: Do not rely on a single executor. Prefer designs that
    leverage BOTH executors creatively within the same workflow
    (e.g., draft with 4o-mini -> verify/refine with Haiku; brainstorm
    with Haiku -> select/format with 4o-mini; parallel
    generate-and-vote using both, etc.).
- If you decide to use only one executor for a step, briefly justify
    why the other is not helpful for that specific step (cost/perf
    tradeoff). Over the whole workflow, ensure both executors have
    meaningful roles.
- Keep each round's modification minimal. Maintain imports to allowed
    modules and record intermediate steps in 'process'.
- Prompts you generate still belong to the 'prompt_custom' module and
    are used by Custom operators regardless of which executor calls
    them.
"""
```

### A.7 DETAILS ABOUT THE DATASETS

**Dataset Selection and Evaluation Protocol.** We use the same six datasets as AFlow (Zhang et al., 2025e): GSM8K, HumanEval, MBPP, HotpotQA, DROP, and MATH. The number of testing samples in each dataset are: HumanEval (131), MBPP (341), MATH (486), GSM8K (1055), HotpotQA (800), and DROP (800). We also evaluate on TravelPlanner (Xie et al., 2024), a benchmark for real-world planning and tool use with Language Agents. The number of testing samples in TravelPlanner are: TravelPlanner (180).

**Shopping-MMLU: Domain-Specific Evaluation.** To assess performance on domain-specific reasoning tasks, we introduce evaluation on Shopping-MMLU, a specialized benchmark for e-commerce reasoning capabilities. Our evaluation protocol for Shopping-MMLU follows a rigorous difficulty-based selection process. We initially evaluated all 33 shopping-related multiple-choice questions using Claude 3.5 Sonnet as a screening model. From these initial evaluations, we identified and selected the 10 most challenging tasks where Claude 3.5 Sonnet achieved less than 80% accuracy, ensuring our evaluation focuses on genuinely difficult reasoning scenarios that require sophisticated multi-agent collaboration.

The selected Shopping-MMLU tasks span critical e-commerce domains including product selection, sentiment analysis, attribute reasoning, multilingual query understanding, product complementarity, semantic classification, brand relationships, keyword intent analysis, review helpfulness assessment, and sentiment evaluation. This comprehensive coverage allows us to evaluate how well different workflow approaches handle the nuanced reasoning required in real-world shopping scenarios. The results for Shopping-MMLU evaluation are presented in Table 7, demonstrating the effectiveness of our approach across diverse shopping-specific reasoning tasks.

### A.8 MORE RESULTS ON THE COST ANALYSIS.

Table 9: Executor-specific token usage and heterogeneous baseline on public benchmarks. Left: input tokens; right: output tokens. Values are averaged over three runs.

| Method (Executor) | Input tokens | | | | | | Output tokens | | | | | |
|---|---|---|---|---|---|---|---|---|---|---|---|---|
| | HumanEval | MBPP | GSM8K | MATH | HotpotQA | DROP | HumanEval | MBPP | GSM8K | MATH | HotpotQA | DROP |
| *GPT-4o mini-based* | | | | | | | | | | | | |
| AFlow (GPT-4o mini) | 2863 | 2049 | 2738 | 10793 | 8501 | 4003 | 1880 | 1409 | 1107 | 5361 | 871 | 605 |
| OneFlow (GPT-4o mini) | 488 | 586 | 1062 | 2096 | 1559 | 1191 | 205 | 196 | 719 | 2286 | 190 | 374 |
| AFlow (GPT-4o mini, single-agent) | 2597 | 414 | 1044 | 9382 | 5572 | 1084 | 1875 | 1279 | 840 | 4979 | 655 | 447 |
| OneFlow (GPT-4o mini, single-agent) | 313 | 443 | 215 | 2164 | 1559 | 1134 | 174 | 196 | 557 | 1782 | 190 | 307 |

Table 10: Executor-specific inference cost (USD) and heterogeneous baseline on public benchmarks. Values are mean ± std over three runs (lower is better).

| Method (Executor) | CODE | | MATH | | QA | |
|---|---|---|---|---|---|---|
| | HumanEval | MBPP | GSM8K | MATH | HotpotQA | DROP |
| *GPT-4o mini-based* | | | | | | |
| AFlow (GPT-4o mini) | $0.198 ± 0.003 | $0.393 ± 0.002 | $1.134 ± 0.001 | $2.343 ± 0.036 | $1.438 ± 0.000 | $0.771 ± 0.001 |
| OneFlow (GPT-4o mini) | $0.026 ± 0.000 | $0.070 ± 0.005 | $0.623 ± 0.001 | $0.819 ± 0.007 | **$0.278 ± 0.000** | $0.322 ± 0.000 |
| AFlow (GPT-4o mini, single-agent) | $0.198 ± 0.004 | $0.283 ± 0.001 | $0.697 ± 0.001 | $2.039 ± 0.028 | $0.530 ± 0.001 | $0.345 ± 0.001 |
| OneFlow (GPT-4o mini, single-agent) | **$0.020 ± 0.000** | **$0.063 ± 0.004** | **$0.387 ± 0.000** | **$0.677 ± 0.002** | **$0.278 ± 0.000** | **$0.284 ± 0.001** |
| *Claude 3.5 Haiku-based* | | | | | | |
| AFlow (Claude 3.5 Haiku) | $0.200 ± 0.000 | $2.390 ± 0.000 | $6.680 ± 0.000 | $3.550 ± 0.000 | $6.550 ± 0.000 | $3.460 ± 0.000 |
| OneFlow (Claude 3.5 Haiku) | $0.250 ± 0.000 | $0.580 ± 0.000 | $5.830 ± 0.000 | $4.330 ± 0.000 | $1.470 ± 0.000 | $3.140 ± 0.000 |
| *Heterogeneous baseline* | | | | | | |
| AFlow (Heterogeneous: GPT-4o mini + Claude 3.5 Haiku) | $0.278 ± 0.003 | $0.343 ± 0.004 | $1.469 ± 0.003 | $1.334 ± 0.006 | $2.153 ± 0.005 | $0.822 ± 0.001 |

