# OpenReview forum: "Rethinking the Value of Multi-Agent Workflow: A Strong Single Agent Baseline"
_ICLR.cc/2026/Conference — Submitted to ICLR 2026_

### Official Review · Reviewer_Ldvv · 2025-10-31

**Soundness:** 2
**Presentation:** 2
**Contribution:** 2
**Rating:** 4
**Confidence:** 2

**Summary:**

This paper examines the value proposition of LLM-based multi-agent systems in settings where current frameworks are largely homogeneous. The authors empirically show that a single LLM using multi-turn conversation with KV cache reuse can match or outperform such multi-agent workflows in both performance and cost across six benchmarks spanning coding, mathematics, and question answering. Building on this finding, they propose OneFlow, an algorithm for automatic, cost-aware workflow optimization tailored for single-agent execution without compromising accuracy.

**Strengths:**

1. The paper offers a reassessment of the prevailing practice of homogeneous multi-agent workflows in LLM systems, combining theoretical reasoning with strong empirical evidence. This provides an important sanity check for the rapidly growing MAS literature.

2. Experiments on six standard and one domain-specific dataset using multiple LLMs convincingly show that single-agent execution can match or surpass homogeneous multi-agent performance while reducing cost substantially.

**Weaknesses:**

1. The heterogeneous experiments (Table 3) rely on automatically generated workflows with unclear tuning and no ablation of model-assignment policies. Thus, the claim that a single-LLM implementation can outperform heterogeneous setups is only provisional.

2. The OneFlow search process is fixed and shallow, with no analysis of sensitivity to search depth, hyperparameters, or model choice. This leaves the robustness of the optimization procedure underexplored.

3. While quantitative results are comprehensive, there is little discussion of failure cases or qualitative differences between single-agent and true multi-agent behaviors, leaving interpretability and diagnostic insight limited.

**Questions:**

1. How does OneFlow’s workflow quality and cost-performance trade-off scale with deeper or wider search? Is the dual meta-LLM architecture robust to prompt or model changes?

2. Have the reported KV-cache efficiency gains been validated using open-weight models that support cache reuse, or might the simulated API-based estimates introduce systematic bias?

---

> ### Author Response · Authors · 2025-11-23
>
> Thank you for your constructive feedback and for finding our discussion on homogeneous multi-agent workflows helpful for the MAS literature. We appreciate your thoughtful comments, which have helped us strengthen the work.
>
> **(1) Clarification of our core claim.**
> Our central argument is that for predefined homogeneous multi-agent workflows where agents collaboratively solve problems, a single LLM agent can execute the original workflow through multi-turn conversation without compromising accuracy. The key insight is that the single agent can reuse a shared KV cache across agent interactions, retaining context without additional prefill cost. This potentially offers efficiency gains and greater consistency than maintaining separate model instances for each agent. Our experiments on standard benchmarks (using AFlow as a baseline) with multiple LLMs (GPT-4o-mini and Claude 3.5 Haiku) were designed to test this claim.
>
> **(2) KV Cache efficiency gains via open-weight LLMs.** To address your concern about validating KV cache efficiency with open-weight LLMs, we conducted additional experiments using Qwen-3-8B with vLLM. Our previous results used proprietary LLMs (GPT-4o-mini with 128k context and Claude 3.5 Haiku with 200k context), which are popular in multi-agent workflow research. We fully agree that open-weight LLMs are highly relevant to this discussion. We selected Qwen-3-8B, one of the state-of-the-art ~8B models, and set the context window to 16k given computing constraints.
>
> The table below shows results on HumanEval, with all benchmarked methods using Qwen-3-8B under the settings described above. Beyond pass@1, we report metrics particularly meaningful for open-weight LLMs: average latency per sample and throughput (samples per second). We highlight the benefits of single-agent execution for both AFlow and OneFlow compared to their original multiple stateless API calls.
> (scores are averaged over three independent runs.)
>
> | Method                             | pass@1 | Avg Latency (s) | Throughput (samples per sec) | Avg Input Tokens | Avg Output Tokens |
> |------------------------------------|--------|------------------|-------------------------------|-------------------|--------------------|
> | CoT      | 83.5% | 2.60    | 7.66  | 213  | 74  |
> | CoT SC × 5       | 83.7% | 17.44   | 1.32 | 1748| 502   |
> | MultiPersona    | 84.7% | 32.38  | 0.75  | 1722  | 846     |
> | AFlow (multiple stateless api calls) | 86.8% | 54.98   | 0.17| 2302 | 1753 |
> | AFlow (Single-agent execution)     | 90.5% | 53.53  | 0.18 | 3269| 1739  |
> | OneFlow (multiple stateless api calls) | 87.0% | 4.31  | 2.47 | 920  | 148|
> | OneFlow (Single-agent execution)   | 87.4% | 4.83   | 1.70 | 1288  | 159|
>
> The main takeaway from GPT-4o-mini and Claude 3.5 Haiku experiments holds: single-agent execution through multi-turn conversation maintains performance (pass@1) and even shows improvement with Qwen-3-8B. Note that for single-agent execution, average input tokens increase (+967 for AFlow and +368 for OneFlow) because multi-turn conversation includes previous conversation history. However, thanks to KV cache, inference efficiency (average latency and throughput) remains comparable.
>
> We observed one difference: with GPT-4o-mini and Claude 3.5 Haiku, single-agent execution with multi-turn conversation reduced total tokens compared to multiple stateless API calls, as models tended to be more concise during multi-turn conversation. With Qwen-3-8B, the multi-turn conversation generated longer responses than single-turn conversation, which explains why we do not observe the same efficiency gains as with GPT-4o-mini.
>
>
>
> **(3) OneFlow design, sensitivity, and goals.**  Our primary goal is to validate that under homogeneous settings, predefined multi-agent workflows can be effectively simulated by a single LLM. The key insight is that task decomposition matters most, rather than the number of agents. Our motivation for designing OneFlow is to automatically create multi-agent workflows that are more streamlined and easier for single-LLM implementation when base models are identical (homogeneous). The designer LLM explores workflow creativity, while the critic LLM implements the proposed workflow in a more cost-effective and streamlined manner. We found that single-LLM implemented OneFlow matches or approaches the performance of multi-agent workflows with substantially greater cost-effectiveness.
>
> For MCTS hyperparameter settings, we followed AFlow (e.g., number of iteration rounds). Empirically, search depth (number of iterations) beyond 20 contributes little to final performance gain, so we set the number of iterations to 20 following AFlow. Both AFlow and OneFlow, along with their single-agent execution variants, support our main claim: for homogeneous LLMs, single-LLM agent simulated workflows are effective.
>
> We look forward to continuing the discussion on this important topic and are happy to provide any additional clarifications or experiments.

---

> > ### Comment · Reviewer_Ldvv · 2025-11-26
> >
> > Thank you for the rebuttal. The authors have clarified the key concerns I raised. I am raised my score.

---

> > > ### Author Response · Authors · 2025-11-26
> > >
> > > Dear Reviewer Ldvv,
> > >
> > > Thank you very much for your careful review and positive feedback, which have helped us improve our work significantly. We really appreciate your support and the time you have taken to engage with our rebuttal.
> > >
> > > Best regards,
> > >
> > > The Authors

---

### Official Review · Reviewer_b2kC · 2025-11-01

**Soundness:** 2
**Presentation:** 2
**Contribution:** 3
**Rating:** 4
**Confidence:** 3

**Summary:**

The paper shows that homogeneous MAS workflows (same base LLM, different prompts/tools) can be simulated by a single LLM via multi-turn role-play. Then it proposes OneFlow, which consists of two parts: 1. search for optimized workflow. 2. perform single LLM implementation. across six benchmarks, single-agent execution often matches or slightly exceeds multi-agent performance but the price is much cheaper.

**Strengths:**

* The paper proposes an interesting point of view.
* The experiments test on six benchmark and report both accuracy and cost to support claims.

**Weaknesses:**

* The OneFlow methods composes of two parts: search for optimized workflow and perform single LLM implementation. The first part seems like an improved version of Aflow and lacks novelty, for example, the critic prompt is adopted from AFlow.
* The costs for single-agent are simulated due to closed-weight APIs; add open-weight runs (or vendor KV-sharing APIs) to validate real-world latency/$ savings
* While the method mentions tool calling, the benchmark tested are static QA/math/code; include tool-use tasks with external side-effects and interactive settings.

**Questions:**

* See weakness.
* Clarification: In 4.2 single-LLM simulator, it writes "Set the system message to p_{i_t}", does this mean to replace the system prompt at the beginning? Can you give an example of how this is different from multi-agent system?

---

> ### Author Response · Authors · 2025-11-23
>
> Thank you very much for finding this work's contribution useful and providing constructive feedback.
>
> **Our main claim and the design of OneFlow.** Our main claim is that when the base model is homogeneous, task decomposition is more important than the formal number of agents in the workflow. AFlow is one of the first and best automatic agentic workflow design frameworks, utilizing a single meta-LLM to iteratively optimize workflows using MCTS. OneFlow automatically designs creative and streamlined multi-agent workflows that are more suitable for conversion to single-agent implementation. During workflow discovery, OneFlow is largely inspired by AFlow with improvements to meta-LLM (critic & designer) communication and collaboration. The second stage is why it is called “OneFlow”: we can use one single LLM to implement it. For AFlow, we also find that its searched workflows can be converted to single-LLM implemented workflows with gains from KV cache, aligning with our main claim.
>
>
> **Real-world latency and performance of single-LLM implemented multi-agent workflows using Qwen3 8B.** The previous results for single-LLM based multi-agent workflow execution were on two proprietary LLMs: GPT-4o-mini (128k context) and Claude 3.5 Haiku (200k context). Though these are popular in multi-agent workflow design research, we totally agree that open-weight LLMs are essential to this discussion. We chose one of the state-of-the-art ~8B models, Qwen-3-8B with vLLM. We set the context window to 16k considering computing resources.
>
> We show the results on HumanEval in the table below. All benchmarked methods use Qwen-3-8B with the above settings. Besides reporting pass@1, we also report metrics that are more meaningful for open-weight LLMs: average latency per sample and throughput (samples per second). We highlight the benefits of single-agent executed AFlow and OneFlow compared to their original multiple stateless API calls. The main takeaway from GPT-4o-mini and Claude 3.5 Haiku holds: using single-agent execution through multi-turn conversation maintains performance (pass@1) and even increases it with Qwen-3-8B. We note that for single-agent execution, the average input tokens increase (+967 for AFlow and +368 for OneFlow) because multi-turn conversations technically require passing previous conversation history to the LLM again, increasing input token length. However, thanks to KV cache, the inference efficiency (average latency and throughput) does not change significantly.
>
> One difference we found is that when experimenting with GPT-4o-mini and Claude 3.5 Haiku, compared to multiple stateless API calls, single-agent executed multi-agent workflows with multi-turn conversations can reduce the total number of tokens and contribute significantly to efficiency (i.e., the model tends to speak more concisely during multi-turn conversations). For Qwen-3-8B, multi-turn conversations tend to produce longer outputs than single-turn conversations. This is why we do not see the same efficiency increase with Qwen-3-8B implemented workflows as we do with GPT-4o-mini implemented workflows.
>
>
> **Tool-use tasks with external side-effects and interactive settings.** For both predefined agentic workflows (like in AFlow and OneFlow) and autonomous agents (e.g., ReAct, Chain-of-Agent), integrated tools significantly aid reasoning. In our previous experiments, we included one tool: a Python sandbox that agents can use to run Python code on mathematical tasks (MATH, GSM8K) and coding tasks (HumanEval, MBPP). This Python sandbox proves very helpful for multi-agent workflows, and single-LLM implemented multi-agent workflows can also use this sandbox effectively. We believe testing additional tool usage abilities, such as web search and file readers, on relevant benchmarks will strengthen our claim about the benefits of single-LLM implemented multi-agent workflows. One consideration is that for tools returning long context from the environment (e.g., Wikipedia MCP or web search), single-agent implementation with LLMs having shorter context windows is more prone to failure.
>
> **For the clarification question in section 4.2:** Thank you very much for bringing this up. It uses a single agent with multi-turn conversations to mimic the multi-agent workflow. We append the system message p_it to the end of the conversation history h_t-1 (we can effectively treat the 'system message' in a multi-agent system as a user message in the single-agent simulated multi-agent system), keeping all previous context. This retains context without additional prefill cost (due to kv-cache) and potentially offers greater consistency than maintaining separate model instances for each agent. We will update the paper to make this point clearer.
>
> Thank you again for your thoughtful and constructive comments. Looking forward to continuing the conversation!

---

> > ### Author Response · Authors · 2025-11-23
> >
> > (scores are averaged over three independent runs.)
> >
> > | Method                             | pass@1 | Avg Latency (s) | Throughput (samples per sec) | Avg Input Tokens | Avg Output Tokens |
> > |------------------------------------|--------|------------------|-------------------------------|-------------------|--------------------|
> > | CoT                                | 83.5% | 2.60             | 7.66                          | 213               | 74                 |
> > | CoT SC × 5                         | 83.7% | 17.44            | 1.32                          | 1748              | 502                |
> > | MultiPersona                       | 84.7% | 32.38            | 0.75                          | 1722              | 846                |
> > | AFlow (multiple stateless api calls) | 86.8% | 54.98            | 0.17                          | 2302              | 1753               |
> > | AFlow (Single-agent execution)     | 90.5% | 53.53            | 0.18                          | 3269              | 1739               |
> > | OneFlow (multiple stateless api calls) | 87.0% | 4.31             | 2.47                          | 920               | 148                |
> > | OneFlow (Single-agent execution)   | 87.4% | 4.83             | 1.70                          | 1288              | 159                |

---

> > > ### Comment · Reviewer_b2kC · 2025-11-25
> > > **Reply**
> > >
> > > Thank you for your replies.
> > >
> > > In your added experiments, it seems that agents with api calls have lower accuracy, which is counter-intuitive that using tools can improve performance.
> > >
> > > Also why AFlow have significantly more output tokens? As you explained, the difference of you method is to append the instruction to the end of the conversation, is this the reason?

---

> > > > ### Author Response · Authors · 2025-11-25
> > > >
> > > > Dear Reviewer b2kC,
> > > >
> > > > Thank you very much for your follow-up questions. We appreciate the opportunity to discuss these points further!
> > > >
> > > > Regarding your question 1: "In your added experiments, it seems that agents with api calls have lower accuracy, which is counter-intuitive that using tools can improve performance."
> > > >
> > > > Yes, adding tools (e.g., a Python sandbox or a web search API) does help boost performance. For example, in the HumanEval task (see the attached table and Table 4 in the updated PDF), the final four methods: (1) AFlow (multiple stateless API calls), (2) AFlow (single-agent execution), (3) OneFlow (multiple stateless API calls), and (4) OneFlow (single-agent execution) all utilize a Python interpreter as a tool and achieve better performance compared to baseline non-tool methods. This is also consistent with the GPT-4o-mini results.
> > > >
> > > > We want to clarify that "AFlow (multiple stateless API calls)" refers to the homogeneous multi-agent workflow optimized with the AFlow framework, which includes multiple stateless LLM API calls (in this example, Qwen3-8B API set up on our local machine). "AFlow (single-agent execution)" means we take the original multi-agent workflow searched by AFlow and execute it using a single Qwen3-8B instance through multi-turn conversations. Even with an open-weight LLM (Qwen3-8B), we observe that the performance matches or even exceeds that of AFlow (multiple stateless API calls), with similar efficiency. Although the performance and efficiency gains with Qwen3-8B are not as pronounced as with more advanced closed-source LLMs (GPT-4o-mini and Claude 3.5 Haiku), the results are consistent with our main claim: for homogeneous multi-agent workflows on collaborative tasks, a single LLM can effectively mimic the multi-agent workflow using multi-turn conversations.
> > > >
> > > >
> > > > Regarding your question 2: Why does AFlow have significantly more output tokens?
> > > >
> > > > That's an excellent observation. We want to clarify: "AFlow (multiple stateless API calls)" is the framework proposed in the AFlow paper. Here, we implemented it by appending instructions to the end of the conversation through multi-turn conversations, resulting in "AFlow (single-agent execution)," which actually shows better performance than the original AFlow (multiple stateless API calls) when using Qwen3-8B.
> > > >
> > > > Compared to AFlow, OneFlow has an additional 'critic' meta-LLM that searches for more streamlined multi-agent workflows. This is why its output tokens are fewer in the table, for both Qwen3-8B and the two closed-source LLM APIs (GPT-4o-mini and Claude 3.5 Haiku). Additionally, with the two closed-source LLM APIs, we found that OneFlow's performance was even better despite using fewer tokens.
> > > >
> > > > In summary, we want to highlight that for multi-agent workflows, what matters is how the task is decomposed, not the number of agents in a specific workflow. Especially for homogeneous multi-agent workflows where the base model is the same, a single LLM simulating the multi-agent workflow with multi-turn conversations can be as effective as multi-agent systems while achieving better efficiency. We hope our work contributes the perspective that for related multi-agent system research on collaborative tasks, researchers should consider an important question: "Do we really need multi-agents, or can we use a single agent with multi-turn conversations to solve the task more efficiently?" We believe this perspective should be considered as an important baseline for future work.
> > > >
> > > > We sincerely appreciate your insightful questions and feedback, which have helped us better clarify our contributions. Thank you for your careful review and dedication!
> > > >
> > > > Best regards,
> > > >
> > > > The Authors

---

> > > > > ### Author Response · Authors · 2025-12-01
> > > > >
> > > > > Dear Reviewer b2kC,
> > > > >
> > > > > Thank you for your follow-up questions regarding tool-use settings. Beyond the original Python interpreter, we have added **TravelPlanner**, a tool-intensive benchmark with long context requirements that necessitates handling multiple constraints and outputs from external tools (e.g., CitySearch, FlightSearch, DistanceMatrix, RestaurantSearch). This benchmark further validates the efficiency gains of single-LLM-executed multi-agent workflows in realistic, complex tool-use scenarios.
> > > > >
> > > > >
> > > > > **Additional results on tool-intensive benchmarks**
> > > > >
> > > > > Results on **TravelPlanner** (https://osu-nlp-group.github.io/TravelPlanner/). All values are averaged over three independent runs.
> > > > >
> > > > > | Method | Performance (±STD) | Cost (±STD) | Avg Input Tokens | Avg Output Tokens |
> > > > > |--------|-------------------|-------------|--------------|---------------|
> > > > > | IO | 36.0% ± 0.1% | 0.307 ± 0.000 | 9,660 | 426 |
> > > > > | AFlow (multiple stateless API calls) | 45.3% ± 0.4% | 2.797 ± 0.023 | 83,366 | 5,057 |
> > > > > | AFlow (single-agent execution) | 46.1% ± 0.4% | 2.373 ± 0.006 | 66,293 | 5,399 |
> > > > > | OneFlow (multiple stateless API calls) | 59.9% ± 1.0% | 0.778 ± 0.001 | 21,276 | 1,884 |
> > > > > | OneFlow (single-agent execution) | **60.1% ± 0.7%** | **0.490 ± 0.000** | **10,556** | **1,902** |
> > > > >
> > > > > These results on TravelPlanner further validate our main claim: single-LLM execution of multi-agent workflows maintains or improves performance while substantially reducing costs, even in tool-intensive scenarios requiring complex constraint handling and long-context processing.
> > > > >
> > > > > We hope this additional evidence addresses your concerns about broader tool-use settings.
> > > > >
> > > > > Best regards,
> > > > >
> > > > > The Authors

---

### Official Review · Reviewer_VtaY · 2025-11-01

**Soundness:** 3
**Presentation:** 2
**Contribution:** 2
**Rating:** 4
**Confidence:** 4

**Summary:**

This paper investigates whether the advantages of multi-agent systems built from homogeneous LLMs can be replicated by a single LLM through multi-turn interactions and KV-cache sharing. The authors empirically evaluate this hypothesis across six benchmarks (code generation, mathematics, and QA tasks) and introduce OneFlow, an automated workflow design algorithm that employs dual meta-LLMs (Designer and Critic) under an MCTS framework. The results suggest that single-agent implementations can match or exceed the performance of multi-agent workflows while substantially reducing inference cost. The paper further discusses the limits of this equivalence in heterogeneous multi-agent contexts and proposes directions for future research.

**Strengths:**

- **S1.** The paper tackles a timely and important issue whether multi-agent systems provide real advantages over single-agent reasoning when the base LLM is homogeneous.

- **S2.** Well-explained theoretical formulation that logically connects shared KV cache to computational efficiency.

- **S3.** Comprehensive experimental coverage across six benchmarks and one domain-specific dataset.

**Weaknesses:**

- **W1.** The OneFlow framework largely replicates the AFlow architecture with minor adaptations. The use of MCTS for workflow generation is not new, and the manuscript does not clearly articulate what conceptual or technical innovation distinguishes OneFlow from AFlow.

- **W2.** The evaluation primarily relies on closed-weight models (GPT-4o-mini, Claude 3.5 Haiku), and the KV-cache advantages are simulated rather than directly measured. The real experiments using open models capable of genuine KV sharing are absent, limiting the credibility of efficiency claims.

- **W3.** The paper primarily contrasts with AFlow and manual CoT baselines, omitting recent heterogeneous agentic frameworks (e.g., MasRouter) that could reveal where single-agent designs fail.

- **W4.** No exploration of when and why the single-agent execution begins to fail (e.g., under longer reasoning chains or tool dependencies).

**Questions:**

- **Q1.** Could the authors provide concrete evidence (with open-weight models) that KV-cache reuse yields measurable cost savings in practice rather than theoretical simulation?

- **Q2.** How does OneFlow's Designer-Critic interaction differ algorithmically from AFlow's meta-LLM setup beyond re-using prompts?

- **Q3.** Several datasets (e.g., GSM8K, MBPP) are solvable via direct prompting. Have the authors tested tasks that genuinely require multi-stage *agentic* reasoning or tool usage?

---

> ### Author Response · Authors · 2025-11-23
>
> Thank you very much for your valuable insights!
>
> Our main claim is that when the base model is homogeneous, what matters more is task decomposition rather than the formal number of agents in the workflow.
>
> **(1) OneFlow and AFlow distinctions.**
> For both Aflow and OneFlow, our goal is to test the argument: for homogeneous multi-agent workflows, can a single-agent-executed multi-agent workflow be comparable to the multi-agent workflow? AFlow is one of the first automatic agentic workflow design frameworks, utilizing a single meta-LLM to iteratively optimize workflows using MCTS. OneFlow features automatic design of creative and more streamlined multi-agent workflows that are better suited for conversion to single-agent implementation. OneFlow consists of two stages: (1) workflow discovery and (2) conversion of multi-agent workflows into single-agent implemented workflows.
> (1) workflow discovery: OneFlow is largely inspired by AFlow with improvements in two meta-LLM communication and collaboration. Specifically, we encourage the creative Designer to boldly explore creative designs and use the Critic to implement them in a more practical way. We ask the Designer and Critic to communicate through comments in the generated code. In the second stage, which is why it is called “OneFlow,” we use a single LLM to implement the workflow. For AFlow, we also find that its searched workflows can be converted to single LLM implementations with gains from KV cache.
>
> **(2) Evidence of KV cache gains using open-weight LLM: Qwen-3 8B.**
> We fully agree that open-weight LLMs can provide practical insights into KV cache gains in our discussion. We chose Qwen-3 8B, one of the state-of-the-art 8B models, with vLLM and set the context window to 16k. Previously, we found that single LLM implementations of multi-agent workflows can maintain performance while reducing cost with two proprietary LLMs: GPT-4o-mini (128k context) and Claude 3.5 Haiku (200k context). For this open-weight LLM experiment, besides reporting pass@1, we also report metrics that are more meaningful for open-weight LLMs: average latency per sample and throughput (samples per second).
>
> As shown in the HumanEval dataset (table below), we highlight the benefits of single-agent executed AFlow and OneFlow compared to their original multiple stateless API calls. The main takeaway from GPT-4o-mini and Claude 3.5 Haiku holds: using single-agent execution through multi-turn conversations maintains performance (pass@1), and even increases it with Qwen-3 8B. We note that for single-agent execution, the average input tokens increase (+967 for AFlow and +368 for OneFlow) because multi-turn conversations technically pass the previous conversation history to the LLM again, increasing input token length. However, thanks to KV cache, the inference efficiency (average latency and throughput) does not change significantly.
>
> One difference we found is that when experimenting with GPT-4o-mini and Claude 3.5 Haiku, compared to multiple stateless API calls, a single-agent executing a multi-agent workflow with multi-turn conversations can reduce the total number of tokens and contribute significantly to efficiency (i.e., the model tends to speak more concisely during multi-turn conversations). For Qwen-3 8B, the multi-turn conversations tend to be longer than single-turn conversations. This is why we do not see the same large efficiency increase in Qwen-3 8B implemented workflows that GPT-4o-mini implemented workflows show.
>
> | Method  | pass@1 | Avg Latency (s) | Throughput (samples per sec) | Avg Input Tokens | Avg Output Tokens |
> |---|-----|-----|-----|------|-----|
> | AFlow | 86.8% | 54.98 | 0.17  | 2302 | 1753  |
> | AFlow (Single-agent)     | 90.5% | 53.53  | 0.18  | 3269 | 1739  |
> | OneFlow | 87.0% | 4.31| 2.47  | 920  | 148   |
> | OneFlow (Single-agent)   | 87.4% | 4.83  | 1.70  | 1288   | 159  |
>
> **(3) Benchmarks requiring higher tool use ability.**
> Thank you for your valuable thoughts. In the previous experiments, we included one tool: a Python sandbox that the agent can use to run Python code on mathematical (MATH, GSM8K) and coding tasks (HumanEval, MBPP). This Python sandbox proves very helpful on these tasks for multi-agent workflows, and single LLM implementations of multi-agent workflows can also use this Python sandbox effectively. We believe testing benchmarks with more diverse tool usage abilities (e.g., web search, file readers) will make our claim about the benefits of single LLM implemented multi-agent workflows stronger. One consideration is that for tools that return long context from the environment (e.g., Wikipedia MCP or web search), LLMs with shorter context windows are more likely to fail when implemented as a single agent.
>
> Thank you for your comments and questions! We look forward to continuing the discussion and hearing your thoughts on our responses.

---

> > ### Comment · Reviewer_VtaY · 2025-11-26
> >
> > Thank you for the detailed rebuttal. The additional experiments with Qwen-3 8B provide some empirical support for KV-cache efficiency, though the evidence remains limited and does not fully resolve concerns about practical gains across diverse open-weight models (or different sizes). The clarification on the distinction between OneFlow and AFlow is helpful but still feels incremental, as the conceptual differences remain modest relative to prior work. The missing comparisons to heterogeneous multi-agent frameworks and broader tool-use settings remain important gaps for assessing the generality of the central claim. Overall, while the rebuttal improves clarity, it does not substantially change my assessment of the paper's novelty or the strength of the empirical validation. I am likely to keep my score unchanged.

---

> > > ### Author Response · Authors · 2025-12-01
> > >
> > > Dear Reviewer VtaY,
> > >
> > > Thank you very much for your continued engagement in this discussion. We greatly appreciate your thoughtful feedback and have conducted additional experiments to address the remaining concerns about tool-use breadth and practical validation.
> > >
> > > To address the concern about broader tool-use settings, we have added **TravelPlanner**, a tool-intensive and context-intensive benchmark that requires handling multiple constraints and outputs from external tools (e.g., CitySearch, FlightSearch, DistanceMatrix, RestaurantSearch). This benchmark further validates the efficiency gains of single-LLM-executed multi-agent workflows in realistic, complex tool-use scenarios.
> > >
> > > **Additional results on tool-intensive benchmarks**
> > >
> > > Results on **TravelPlanner** (https://osu-nlp-group.github.io/TravelPlanner/). All values are averaged over three independent runs.
> > >
> > > | Method | Performance (±STD) | Cost (±STD) | Avg Input Tokens | Avg Output Tokens |
> > > |--------|-------------------|-------------|--------------|---------------|
> > > | IO | 36.0% ± 0.1% | 0.307 ± 0.000 | 9,660 | 426 |
> > > | AFlow (multiple stateless API calls) | 45.3% ± 0.4% | 2.797 ± 0.023 | 83,366 | 5,057 |
> > > | AFlow (single-agent execution) | 46.1% ± 0.4% | 2.373 ± 0.006 | 66,293 | 5,399 |
> > > | OneFlow (multiple stateless API calls) | 59.9% ± 1.0% | 0.778 ± 0.001 | 21,276 | 1,884 |
> > > | OneFlow (single-agent execution) | **60.1% ± 0.7%** | **0.490 ± 0.000** | **10,556** | **1,902** |
> > >
> > > **Clarification on scope and contributions**
> > >
> > > We acknowledge that establishing when and why heterogeneous multi-agent workflows become necessary remains an open question. Our work provides a strong single-agent execution baseline that can serve as a foundation for future research to evaluate when true heterogeneity is required. We have updated the manuscript to better emphasize this positioning: rethinking the value of homogeneous multi-agent workflows through rigorous single-agent baselines, while explicitly calling for more research on heterogeneous multi-agent systems.
> > >
> > > We sincerely appreciate your constructive feedback throughout this process and hope these additional results and clarifications address your concerns.
> > >
> > > Best regards,
> > >
> > > The Authors

---

### Official Review · Reviewer_gFZd · 2025-11-05

**Soundness:** 2
**Presentation:** 3
**Contribution:** 2
**Rating:** 6
**Confidence:** 4

**Summary:**

This paper questions whether multi-agent LLM workflows truly outperform single LLMs when all agents share the same base model.
It formally shows that for homogeneous workflows (same base LLM, different prompts/tools), a single LLM can simulate the entire multi-agent pipeline through multi-turn dialogue while reusing the KV cache, gaining efficiency without loss of expressivity.
Based on this, the authors propose OneFlow, an automatic workflow design framework using dual meta-LLMs (Designer + Critic) and Monte-Carlo Tree Search to generate workflows optimized for single-agent execution.
Across six benchmarks (HumanEval, MBPP, GSM8K, MATH, HotpotQA, DROP) and one domain-specific Shopping-MMLU set, OneFlow-single achieves comparable or better performance than existing multi-agent frameworks (AFlow, etc.) while cutting inference cost by up to 10×.
The paper concludes that homogeneous MAS can be largely simulated by a single agent and that future work should focus on truly heterogeneous systems.

**Strengths:**

The author Reframes multi-agent research with a rigorous single-agent equivalence argument.

they provide Six general benchmarks + domain-specific tasks.

they also Quantifies KV-cache benefits clearly.

it shows that OneFlow’s dual-meta LLM + MCTS is a creative and reproducible design. and it  clearly delineates where single-agent simulation applies and where heterogeneity still matters.

**Weaknesses:**

Limited empirical heterogeneity analysis: Pilot study is small; results inconclusive about real multi-model synergy.

Simulation of KV cache: Since APIs hide internal caching, efficiency results are theoretical. A small open-weight replication (e.g., LLaMA-3 8B) would strengthen credibility.

Ablations: Lack of ablation on MCTS parameters (α, β, iterations) and meta-LLM roles; unclear how much each contributes.

Over-dependence on closed models: Limits reproducibility beyond cost estimation.

Writing could be tighter: Some redundant explanations and long prompts in appendix.

**Questions:**

Can you verify KV-cache reuse gains empirically using an open-source model?

How sensitive are results to the α/β weights in Eq. (1)?

Would OneFlow still outperform AFlow if inference cost were excluded (i.e., pure accuracy metric)?

Have you tested whether role-switching (different prompts within same chat) affects coherence or context interference?

How does OneFlow perform when the base model has small context windows (e.g., 4k tokens)—does summarization degrade accuracy?

---

> ### Author Response · Authors · 2025-11-23
>
> Thank you very much for your constructive feedback and insights.
>
> Our main claim is that for predefined homogeneous multi-agent workflows where each agent collaboratively solves problems, the base LLM's homogeneity allows us to employ a single LLM agent to execute the original multi-agent workflow through multi-turn conversation. Our experiments show this can serve as a strong baseline when the multi-agent workflow is homogeneous.
>
> **(1) KV cache reuse with open-weight LLM: Qwen-3 8B.**
> We agree that open-weight LLMs are essential for this discussion. While our previous results used two proprietary LLMs (GPT-4o-mini with 128k context length and Claude 3.5 Haiku with 200k context length), which are popular in multi-agent workflow design research, we have now conducted additional experiments using Qwen-3 8B with vLLM. We set the context window to 16k considering available computing resources, which also allows us to examine how context window size influences performance.
>
> The table below shows results on HumanEval, with all benchmarked methods using Qwen-3 8B under the same settings. Besides pass@1, we report metrics that are more meaningful for open-weight LLMs: average latency per sample and throughput (samples per second). We highlight the benefits of single-agent execution for both AFlow and OneFlow compared to their original multiple stateless API calls. The main takeaway from GPT-4o-mini and Claude 3.5 Haiku holds: single-agent execution through multi-turn conversation maintains performance (pass@1) and even increases it with Qwen-3 8B. We observe that for single-agent execution, average input tokens increase (+967 for AFlow and +368 for OneFlow). This is because multi-turn conversation technically gives the previous conversation history to the LLM again, increasing input token length. Thanks to KV cache, inference efficiency (average latency and throughput) does not change substantially.
>
> One difference we found is that when experimenting with GPT-4o-mini and Claude 3.5 Haiku, single-agent execution with multi-turn conversation reduced the total number of tokens compared to multiple stateless API calls, contributing significantly to efficiency (the model tends to speak more concisely during multi-turn conversation). For Qwen-3 8B, the multi-turn conversation tends to generate longer responses than single-turn conversation. This is why we do not see as large an efficiency increase with Qwen-3 8B as we observed with GPT-4o-mini.
>
> (scores are averaged over three runs.)
>
> | Method  | pass@1 | Avg Latency (s) | Throughput (samples per sec) | Avg Input Tokens | Avg Output Tokens |
> |------------------------------------|--------|------------------|-------------------------------|-------------------|--------------------|
> | CoT  | 83.5% | 2.60             | 7.66  | 213    | 74    |
> | CoT SC × 5     | 83.7% | 17.44   | 1.32    | 1748              | 502 |
> | MultiPersona    | 84.7% | 32.38            | 0.75   | 1722      | 846 |
> | AFlow| 86.8% | 54.98  | 0.17  | 2302     | 1753   |
> | AFlow (Single-agent)  | 90.5% | 53.53 | 0.18     | 3269   | 1739    |
> | OneFlow | 87.0% | 4.31 | 2.47    | 920   | 148   |
> | OneFlow (Single-agent)   | 87.4% | 4.83| 1.70   | 1288  | 159    |
>
> **(2) OneFlow sensitivity to hyperparameters.**
> OneFlow and Aflow in this work serve for our main claim: when the base model is homogeneous, what matters more is task decomposition rather than the formal number of agents in the multi-agent workflow.. In OneFlow, the designer LLM tries its best to explore bold workflow creativity, while the critic LLM tries its best to implement the designer's proposed workflow in a more cost-effective and streamlined way. We found that single-LLM-implemented OneFlow's (and Aflow) performance can match the performance of multi-agent workflows with greater cost-effectiveness.
>
> For MCTS hyperparameter settings, we followed AFlow (for example, the number of iteration rounds). Empirically, we found that search depth (number of iterations) beyond 20 contributes little to final performance gain. Therefore, we followed AFlow in setting the number of iterations to 20.
>
> **(3) Role switching and multi-turn coherence.**
> For tasks with very distinct roles (for example, in a healthcare-related multi-agent system where one LLM role-plays a patient and another role-plays a doctor), role switching within a single LLM can be too drastic and may not be feasible. However, we argue that for collaborative tasks (for example, agents working together to solve a math problem, coding problem, or QA problem), providing information completeness (context) is very beneficial for solving the problem. In such collaborative settings, role switching within multi-turn conversation does not introduce coherence issues or context interference, as all roles share the common goal of problem-solving.
>
> Thank you again for your thoughtful comments. We appreciate the opportunity to address your concerns and welcome any additional feedback or suggestions.

---

> > ### Author Response · Authors · 2025-12-01
> >
> > Dear Reviewer gFZd,
> >
> > Thank you very much for your feedback! We have added TravelPlanner, a tool-intensive and context-intensive benchmark, to further demonstrate the advantages of single-agent execution for multi-agent workflows. This benchmark requires the agentic workflow to handle multiple constraints and outputs from external tools (e.g., CitySearch, FlightSearch, DistanceMatrix, RestaurantSearch), which further validates the efficiency gains of single-LLM-executed multi-agent workflows.
> >
> > Results on **TravelPlanner** (https://osu-nlp-group.github.io/TravelPlanner/). All values are averaged over three independent runs.
> >
> > | Method | Performance (±STD) | Cost (±STD) | Avg Input Tokens | Avg Output Tokens |
> > |--------|-------------------|-------------|--------------|---------------|
> > | IO | 36.0% ± 0.1% | 0.307 ± 0.000 | 9,660 | 426 |
> > | AFlow (multiple stateless API calls) | 45.3% ± 0.4% | 2.797 ± 0.023 | 83,366 | 5,057 |
> > | AFlow (single-agent execution) | 46.1% ± 0.4% | 2.373 ± 0.006 | 66,293 | 5,399 |
> > | OneFlow (multiple stateless API calls) | 59.9% ± 1.0% | 0.778 ± 0.001 | 21,276 | 1,884 |
> > | OneFlow (single-agent execution) | **60.1% ± 0.7%** | **0.490 ± 0.000** | **10,556** | **1,902** |
> >
> >
> > Regarding the pilot study on heterogeneous multi-agent workflows, we acknowledge that there may exist more efficient, sophisticated manually-crafted multi-agent workflows than automatically-searched ones. Indeed, establishing a strong single-agent execution baseline is one of our key motivations for this work, as it provides a foundation for evaluating when true heterogeneity becomes necessary. We have updated the manuscript to reflect these modifications more clearly.
> >
> > Thank you again for your thorough review and constructive feedback.
> >
> > Best regards,
> >
> > The Authors

---

### Author Response · Authors · 2025-12-01
**Rebuttal Summary**

We sincerely thank all reviewers for their constructive feedback. We have updated the PDF by incorporating reviewer feedback (highlighted in blue font).
## Summary of rebuttal stage and score changes
We appreciate the reviewers finding this work timely and valuable for agentic multi-agent workflow research. Below we summarize the reviewers' concerns and our responses during the rebuttal stage.

---

### **Reviewer Ldvv: 4 → 6**
*(Score raised to 6 before Nov 25 night; concerns addressed)*

**Concerns:**
- Open-weight model experiments on KV-cache reuse
- OneFlow framework's sensitivity and comparison with AFlow
- Provisional conclusions on heterogeneous multi-agent workflows

**Our response:**
- Added experiments using open LLMs (Qwen-3 8B)
- Added experiments on tool-usage benchmark (TravelPlanner)
- Emphasized our main claim and clarified the scope of heterogeneous multi-agent workflow experiments

---

### **Reviewer gFZd: 6**
*(Maintained initial score; discussion stage froze before further interaction)*

**Concerns:**
- Open-source model experiments on KV-cache reuse; over-dependence on closed-weight LLMs
- Provisional conclusions on heterogeneous multi-agent workflows
- OneFlow framework's sensitivity and comparison with AFlow
- Writing clarity

**Our response:**
- Added experiments using open LLMs (Qwen-3 8B)
- Added experiments on tool-usage benchmark (TravelPlanner)
- Emphasized our main claim and clarified heterogeneous workflow experiment goals
- Improved manuscript clarity (updated PDF with blue highlights)

---

### **Reviewer VtaY: 4**
*(Acknowledged open LLM results positively; requested more tool benchmarks; we subsequently added TravelPlanner)*

**Concerns:**
- Open-source model experiments on KV-cache reuse
- Innovation of OneFlow compared to AFlow
- More tool-usage benchmarks
- More heterogeneous multi-agent workflow baselines to address failure cases

**Our response:**
- Added experiments using open LLMs (Qwen-3 8B)
- Added experiments on tool-usage benchmark (TravelPlanner)
- Updated PDF manuscript to emphasize OneFlow's innovation, our main claim, and heterogeneous workflow experiment scope

---

### **Reviewer b2kC: 4**
*(Initial response received Nov 25; follow-up questions asked)*

**Concerns:**
- Innovation of OneFlow compared to AFlow
- Open-weight LLM experiments on KV-cache reuse
- Need for more tool-usage benchmarks

**Our response:**
- Added experiments using open LLMs (Qwen-3 8B)
- Added experiments on tool-usage benchmark (TravelPlanner)
- Updated PDF manuscript to emphasize OneFlow's innovation and our main claim

## Summary of main claims and modifications
Our results demonstrate that for homogeneous multi-agent workflows designed for solving complex tasks (e.g., coding problems, travel planning), **task decomposition is more important than the formal number of agents in the workflow**. We aim to convey the following message to the community: **for homogeneous multi-agent workflows, single-LLM implementation can serve as a more efficient alternative and should be evaluated as a baseline before adopting multi-agent architectures.**

---

**Concern 1: Open-weight LLM experiments** *(Raised by reviewers gFZd, VtaY, b2kC, Ldvv)*

We added experiments using **Qwen-3-8B** on HumanEval, reporting both pass rates and inference metrics (latency, throughput). These results demonstrate **non-trivial performance gains through KV cache reuse** when using single-agent implementation, even with open-weight models.

---

**Concern 2: Tool-usage benchmarks** *(Raised by reviewers VtaY and b2kC)*

We clarified that our original coding and math benchmarks already utilize tools like Python interpreters. We further added experiments on **TravelPlanner**, a tool-intensive benchmark requiring agents to utilize multiple external information sources (e.g., restaurant search, flight search) to generate suitable travel plans. Results show that **single-LLM implementation reduces cost while slightly improving accuracy**, even in these complex tool-use scenarios.

---

**Concern 3: OneFlow's novelty compared to AFlow**
*(Raised by reviewers VtaY and b2kC)*

We elaborated that our primary contribution is demonstrating that **single-agent execution of homogeneous multi-agent workflows serves as a strong baseline, hence the name "OneFlow."** We build upon AFlow's automatic workflow design by introducing a **dual meta-LLM architecture (Designer + Critic)** that collaboratively creates more streamlined multi-agent workflows suitable for single-agent implementation. Our main claim centers on **establishing single-agent implementation as a critical baseline in multi-agent research when the base model is homogeneous**.

---

We have updated the PDF (changes highlighted in blue font) to reflect our efforts in addressing reviewer concerns and better articulating the goals of this work. We deeply appreciate the time and effort the reviewers and AC have invested in this process.

Best regards,

The Authors

---

### Meta-Review · Area_Chair_oswE · 2026-01-07

**Summary:**

Reviewer gFZd views the paper’s main contribution as establishing that homogeneous multi-agent workflows can be simulated by a single LLM via multi-turn interaction, and finds the empirical results generally sound.
Reviewer Ldvv similarly characterizes the work as a sanity check for homogeneous multi-agent systems and acknowledges that the experiments show comparable accuracy with reduced cost, leading them to raise their score after the rebuttal.

In contrast, Reviewers VtaY and b2kC express persistent concerns about the strength of the contribution. Both question whether OneFlow offers a meaningful advance over AFlow.
Reviewer VtaY further emphasizes that the empirical validation of KV-cache efficiency and tool-use generality remains limited, and explicitly states that the rebuttal does not substantially change their assessment.
Reviewer b2kC raises similar concerns about novelty and evaluation scope, without indicating a shift in position after the rebuttal.

**Reviewer Concerns:**

**Addressed**

- KV-cache validation with open-weight models: Added Qwen3 8B latency and throughput experiments address requests from gFZd, VtaY, b2kC, and LdVv.

- Tool-use coverage: Added TravelPlanner partially addresses requests from VtaY and b2kC for more tool-intensive evaluation.

- Clarification of scope: Reframed claims to focus on homogeneous workflows, addressing concerns from gFZd and Ldvv.

**Outstanding**

- Novelty of OneFlow vs. AFlow: VtaY and b2kC maintain that OneFlow remains largely incremental.

- Generality of efficiency gains: VtaY notes evidence is limited to a small set of open-weight models.

- Heterogeneous workflow evidence: gFZd and Ldvv consider the heterogeneous analysis insufficient.

- Search robustness: gFZd and Ldvv note missing sensitivity and ablation analysis.

**Reviewer Scores:**

Reviewer gFZd: Unchanged. They acknowledge the added experiments but maintain a marginal score and state they would not object to rejection.

Reviewer VtaY: Unchanged. They explicitly state the rebuttal does not change their assessment.

Reviewer b2kC: Likely unchanged. The discussion does not indicate their concerns about novelty and scope were resolved.

Reviewer Ldvv: It appears in the revision history that Ldvv has raised from 4 to 6, but still showing 4 as the final score..

---

### Decision · Program_Chairs · 2026-01-26

Reject